# CoMA – an intuitive and user-friendly pipeline for amplicon-sequencing data analysis

**Sebastian Hupfauf** [ID]**[1]\*, Mohammad Etemadi** [ID]**[2], Marina Fernández-Delgado Juárez[1], María Gómez-Brandón[3], Heribert Insam[1], Sabine Marie Podmirseg[1]**

**1** Department of Microbiology, University of Innsbruck, Innsbruck, Austria, **2** Department of Horticultural Science, School of Agriculture, Shiraz University, Shiraz, Iran, **3** Department of Ecology and Animal Biology, GEA Group, University of Vigo, Vigo, Spain

\* Sebastian.Hupfauf@uibk.ac.at

## Abstract

In recent years, there has been a veritable boost in next-generation sequencing (NGS) of gene amplicons in biological and medical studies. Huge amounts of data are produced and need to be analyzed adequately. Various online and offline analysis tools are available; however, most of them require extensive expertise in computer science or bioinformatics, and often a Linux-based operating system. Here, we introduce "CoMA–Comparative Microbiome Analysis" as a free and intuitive analysis pipeline for amplicon-sequencing data, compatible with any common operating system. Moreover, the tool offers various useful services including data pre-processing, quality checking, clustering to operational taxonomic units (OTUs), taxonomic assignment, data post-processing, data visualization, and statistical appraisal. The workflow results in highly esthetic and publication-ready graphics, as well as output files in standardized formats (e.g. tab-delimited OTU-table, BIOM, NEWICK tree) that can be used for more sophisticated analyses. The CoMA output was validated by a benchmark test, using three mock communities with different sample characteristics (primer set, amplicon length, diversity). The performance was compared with that of Mothur, QIIME and QIIME2-DADA2, popular packages for NGS data analysis. Furthermore, the functionality of CoMA is demonstrated on a practical example, investigating microbial communities from three different soils (grassland, forest, swamp). All tools performed well in the benchmark test and were able to reveal the majority of all genera in the mock communities. Also for the soil samples, the results of CoMA were congruent to those of the other pipelines, in particular when looking at the key microbial players.

## Introduction

Nowadays, many studies in environmental microbiology [1, 2], molecular ecology [3, 4] and medical diagnostics [5, 6] are either based on or accompanied by next-generation sequencing (NGS) analyses. Amplicon-sequencing (e.g. 16S rRNA, 18S rRNA, internal transcribed spacer (ITS), functional genes) is a cost- and effort efficient alternative to metagenomic approaches and became a mainstream and high-throughput technique for the determination of biological community structures. The costs have decreased with time and today a sample may be

**Data Availability Statement:** The source code of the CoMA pipeline can be accessed at GitHub (https://github.com/SebH87/coma). The amplicon-sequencing data of the three investigated soils are

available at the European Nucleotide Archive under the accession numbers ERS3378910 to ERS3378921 within study PRJEB32269 (https:// www.ebi.ac.uk/ena/data/view/PRJEB32269).

**Funding:** This work was supported by the Federal Ministry of Education, Science and Research (https://www.bmbwf.gv.at/), and by the University of Innsbruck (https://www.uibk.ac.at/) within the scope of the project "MicrobeEnergy". SH was further supported by the doctoral fellowship (LFU Doktoratsstipendium) of the University of Innsbruck, Austria. MG acknowledges support by the Programa Ramón y Cajal (RYC-2016-21231; Ministerio de Economía y Competitividad, https:// www.mineco.gob.es/). The publication fees are sponsored by the "Fonds zur Förderung der Wissenschaftlichen Forschung" (https://www.fwf. ac.at/; Project I 989). The funders had no role in study design, data collection and analysis, decision to publish, or preparation of the manuscript.

**Competing interests:** The authors have declared that no competing interests exist.

analyzed for less than 50 €. Moreover, amplicon-sequencing allows for the simultaneous screening of hundreds of samples. All these aspects have made amplicon-sequencing approaches extremely popular and led to a multitude of studies investigating all kinds of habitats. Among them there are the microbiota of humans [7], aquatic ecosystems [8], soils [9], deadwood [10], wastewater treatment systems [11] and anaerobic digesters [12–14], to name only a few. Although most of these studies are focused on prokaryotes by targeting the 16S rRNA genes of bacteria and/or archaea, also other structural genes like 18S rRNA [15], 28S rRNA [16, 17] and ITS [18], or different functional genes [19, 20] are commonly used.

However, the downside of these high-throughput approaches are enormous amounts of generated data requiring proper analysis. Modern NGS platforms like Illumina NovaSeq yield up to $2 \times 10^{10}$ sequences per run (https://emea.illumina.com/systems/sequencing-platforms. html), stored in big data files of usually several gigabyte (GB) in size. The typical workflow for amplicon-sequencing data analysis comprises demultiplexing of the dataset in order to assign sequences to samples and merging of paired-end reads to obtain single sequences (however, there are also situations where single-end reads are provided or where merging is not recommended for some reason). Then, trimming off primers, adapter sequences or barcodes and filtering of bad sequences based on quality or sequence length is required. Similar sequences are clustered into operational taxonomic units (OTUs) using a common similarity cutoff (typically 97%) and OTUs are assigned to a taxonomic reference database such as Greengenes [21] (last update 2013), RDP [22] (last update: 2016), SILVA [23], PR2 [24], MIDORI [25], ITSoneDB [26] or UNITE [27], a process called *de novo* assembling (= sequences clustering before taxonomic assignment). The order can also be reversed resulting in a different strategy for OTU picking that is called *closed-reference* or *open-reference* if it is a combination of both approaches. However, these picking strategies are rare nowadays and most tools are using *de novo* assembling. As an alternative to OTUs, current research is going in the direction of amplicon sequence variants (ASVs), where sequences are clustered based on differences down to a single nucleotide [28]. This procedure yields real biological sequences rather than constructed clusters, which can be directly compared between different studies. Moreover, ASV approaches tend to be less time-consuming, and no arbitrary threshold and centroid selections are needed [29]. On the other hand, OTU clustering algorithms such as USEARCH turned out to produce particularly reliable results and these tools are applied by multitudes of scientists. Current publications indicate that both, OTU and ASV approaches yield robust results, which are well comparable to each other [30, 31]. After OTU/ASV picking, data post-processing, including singleton removal and data subsampling may be conducted, followed by statistical appraisal (e.g. alpha-diversity, beta-diversity, hypothesis testing, redundancy analysis) and data visualization.

A wide range of software solutions is available for computing all these steps. Web-based applications promise fast and simple solutions for data analysis. Nonetheless, limited upload capacities and collapsing server speed at peak times often bother the user and may turn online data processing into a frustrating procedure. Beyond web-based applications, there are also various offline tools freely available, among them Mothur [32] and "Quantitative Insights Into Microbial Ecology" (QIIME [33], QIIME2 [34]). Both provide a great variety of different tools in terms of explorative data analysis, quality control (QC), data processing, statistics and data visualization, and allow extensive adjustment and fine-tuning of each step. The drawback of these remarkably wide tool packages and options, however, is that the user needs extensive knowledge and experience on handling command-line tools and sequence-based data processing. This problem can only be partially addressed by the numerously available online documentations and interactive user blogs. If the user does not have sufficient experience and expertise, the complex workflow may be time-consuming and may lead to mistakes and

erroneous conclusions. Moreover, many bioinformatic tools require a Linux operating system and deep operation skills, often lacked by the users, or requiring a time-consuming training. An overview on existing software packages is provided in Table 1 showing the most relevant features as well as drawbacks relative to CoMA.

We developed CoMA (Comparative Microbiome Analysis) as a free pipeline for intuitive and user-friendly analysis of amplicon-sequencing data, available for all common computer

**Table 1. Selection of existing software packages available for amplicon sequencing data analysis.**

| Tool | Important features | Drawbacks relative to CoMA |
|---|---|---|
| AmpliSAT [35] | Set of web-based tools, oriented towards advanced users, flexible usage, accepts various different marker genes | difficult to handle for entry-level users, performance relies on server speed and other job submissions |
| BioMaS [36] | Web-based application, user-friendly design, amplicons from Illumina, Ion Torrent and Roche 454, basic data visualization (pie chart, tax. tree) | no support for PacBio, no post processing, no statistics, plain data visualization without many options, performance relies on server speed and other job submissions |
| CloVR-ITS [37] | optimized for analyzing fungal communities, offers data visualization and statistics, supports cloud computing | limited to ITS primers, limited to Roche 454 → no Illumina data supported |
| ITScan [38] | Web-based application, offers data visualization and statistics, optimized for analyzing fungal communities | limited to ITS primers, performance relies on server speed and other job submissions |
| LotuS [39] | command line-based, more oriented towards advanced users, flexible usage, amplicons from Illumina, Ion Torrent, PacBio and Roche 454 | no GUI, no post processing, no data visualization, no statistics |
| MetaAmp [40] | Web-based application, supports ASVs, moderate complexity, accepts various different marker genes, output in standardized formats | no post processing, no data visualization, performance relies on server speed and other job submissions |
| MG-RAST [41] | Web-based application, also supports other NGS data analysis apart from AS, moderate complexity, flexible usage | performance relies on server speed and other job submissions |
| MICCA [42] | command line-based, single- and paired-end reads, moderate complexity, amplicons from Illumina, Ion Torrent and Roche 454 | no GUI, no post processing, no data visualization, no statistics |
| Microbiome Analyst [43] | Web-based application, also supports other NGS data analysis apart from AS, moderate complexity, flexible usage | performance relies on server speed and other job submissions |
| Mothur [32] | command line-based, oriented towards advanced users, very flexible usage with a huge variety of tools and options | no GUI, difficult to handle for entry-level users, |
| PEMA [31] | user-friendly operation, OTU and ASV supported, provides α- and β-diversity analyses as well as statistics, output in standardized format | currently limited to 4 different marker genes (16S, 18S, ITS, COI), |
| PipeCraft [44] | GUI, user-friendly design, flexible usage, amplicons from Illumina, Ion Torrent, PacBio and Roche 454 | no post processing, no data visualization, no statistics |
| PIPITS [45] | optimized for analyzing fungal communities, extraction of an ITS sub region, uses paired-end reads, amplicons from Illumina | limited to Illumina MiSeq data, limited to ITS primers, no GUI, no post processing, no data visualization, no statistics |
| QIIME [33] | command line-based, oriented towards advanced users, very flexible usage with a huge variety of tools and options | no GUI, requires a Linux OS, difficult to handle for entry-level users |
| QIIME2 [34] | command line-based or GUI-based, oriented towards advanced users, very flexible usage with a huge variety of tools and options, supports ASV | requires a Linux OS, difficult to handle for entry-level users |

platforms. We used various open-source, third-party tools and combined them into a linear analysis workflow in the form of a Bash script, starting with the raw input files (in FASTQ format) and resulting in esthetically pleasing and publication-ready graphics. In addition, output files in standardized formats, such as a tab-delimited OTU-table, an OTU-table in BIOM format and a tree file in NEWICK format, are provided. These allow for subsequent secondary analysis using for example Cytoscape [46], GraPhlAn [47], LEfSe [48], PICRUSt [49] or R [50], if desired. The operation of this tool is remarkably intuitive and makes it accessible even for entry-level users. A graphical user interface facilitates the handling, representing a major advantage compared with command-line based applications. Nevertheless, multiple adjustment parameters and the high degree of automation make CoMA also acceptable for advanced users who are looking for an efficient and streamlined data analysis. The tool is capable of handling data from today's most important NGS platforms, including Illumina MiSeq, Illumina HiSeq, or Illumina NovaSeq, but also from the former 454 pyrosequencing technology, which was, in fact, terminated in 2016 but data are still around and analysis tools are still needed. This study focuses on the processing of short rather than long reads, which are produced by currently emerging third-generation sequencing technologies like PacBio or Nanopore. Appropriate analysis tools for long reads are not discussed here but are readily available (e.g. SDip [51]). Three different options for installation are available for CoMA, which can all be downloaded from the CoMA webpage (https://www.uibk.ac.at/microbiology/services/coma. html). A detailed manual describes both the installation as well as the usage of the tool.

Here, we present the new CoMA pipeline and benchmark it by analyzing three different, constructed mock communities. The performance of CoMA is compared with that of Mothur, QIIME and QIIME2, the currently most popular platforms for NGS data analysis (however, the support for QIIME ended in 2018 and the developers suggest switching to QIIME2). Moreover, we apply CoMA to a real dataset on the prokaryotic microbiota of three different soils (forest, grassland, swamp) in order to demonstrate its functionality also on a practical example. Soil samples were analyzed on an Illumina MiSeq device, using a 16S rRNA amplicon-sequencing approach. We hypothesize that the CoMA pipeline works well in the benchmark test, revealing all included genera. We expect CoMA to perform as efficiently and precisely as Mothur, QIIME and QIIME2, and that the results are stable irrespective of the applied dataset. In addition, we hypothesize that the results for the soil samples obtained by the different analysis tools are highly comparable, especially when looking at the key microbial players.

## Material and methods

### Implementation of CoMA

CoMA is implemented as a linear data analysis pipeline ranging from the processing of raw input files to the computation of results. It comprises four main sections: data pre-processing and quality checking, OTU clustering and taxonomic assignment, data post-processing, as well as data visualization and statistical appraisal (Fig 1). All steps, except for the merging of paired-end reads and OTU clustering/taxonomic assignment, are optional and can be skipped by the user if desired. Therefore, the tool allows for the (partial) re-calculation of a former, already completed CoMA run or the continuation of an unfinished or terminated analysis. CoMA offers a simple user interface, composed of GTK+ dialogs, which guides the user through the analysis and provides important information on each step of the workflow.

**Data pre-processing and quality checking.** Data analysis using CoMA requires input files in FASTQ format (paired-end or single-end, either uncompressed or compressed with *gzip*), which is the standard NGS file format provided by all common sequencing platforms. In case of paired-end reads, forward and reverse reads are first merged at their overlapping

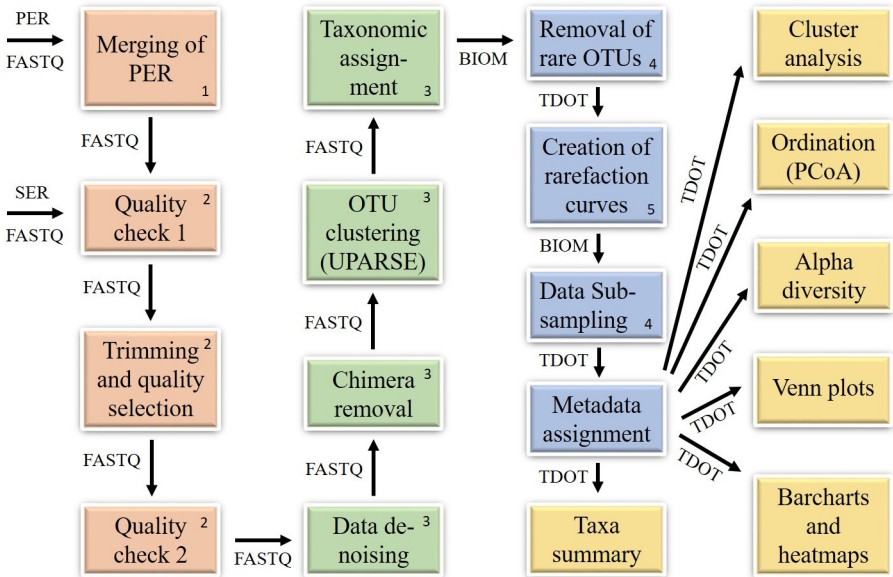

**Fig 1. Overview of the CoMA pipeline workflow.** Different colors represent the four sub-sections of the CoMA workflow: Data pre-processing and quality checking (orange), clustering of operational taxonomic units (OTUs) and taxonomic assignment (green), data post-processing (blue) and data visualization and statistical appraisal (yellow). Labelled arrows demonstrate the order of events and name specific file types that are needed as input for each step. Taxonomic assignment is done with Blast, Lambda or RDP using either one of the available databases (e.g. Silva [23]) or any custom database provided by the user. Numbers indicate third party tools that are used for the specific CoMA step: 1 = PANDAseq, 2 = PRINSEQ, 3 = LotuS/sdm, 4 = QIIME, 5 = Mothur. TDOT = Tab-delimited OTU-table. PER = Paired-end reads. SER = Single-end reads. PCoA = Principal coordinates analysis.

region, using the open-source tool "Paired-end assembler for DNA sequences" (PANDAseq [52]). This step is followed by a QC step (an individual quality report is created for each sample) and a subsequent sequence selection step, both accomplished by the "Preprocessing and Information of Sequence data" (PRINSEQ [53]) utility. Amplification primers, sequencing adapters and multiplex identifiers are removed (trimming) and reads are selected based on Phred quality score (PQS), amplicon length and ambiguous bases. The selection parameters are different for each specific primer pair and can be adjusted based on the information gained from QC. A second QC step, checking the success of trimming and sequence selection, finally completes the data pre-processing workflow.

**OTU clustering and taxonomic classification.** All steps within this program section are either part of the "Less OTU scripts" (LotuS [39]) pipeline or the "simple demultiplexer" tool (sdm [39]). First, an error-correction step is applied, followed by the removal of artificial chimeric sequences, resulting from cross-hybridization of DNA fragments during library preparation. Subsequently, clustering to OTUs is done with the UPARSE-OTU algorithm [54] at a 97% similarity level, following a de-novo assembling approach. OTU sequences are then aligned/classified with a desired gene reference database using either "Basic Local Alignment Search Tool" (BLAST [55]), "Local Aligner for Massive Biological Data" (Lambda [56]) or the "Ribosomal database project" (RDP [57, 58]) classifier. CoMA supports several popular taxonomic databases, including SILVA, Greengenes, RDP, UNITE, HITdb [59], beetax [60], and PR2, as well as any custom database provided by the user. A custom database needs to be composed of two individual files: a FASTA file with all sequences and a TAX file with the accompanying taxonomic information. The correct structure for both files is provided in the CoMA manual. It is possible to use multiple databases, where the first serves as primary- and the

other(s) as backup database(s). OTUs that cannot be assigned to the primary database are then diverted to the backup database(s). This workflow section is completed by the generation of universally applicable output files (tab-delimited OTU-table, BIOM file, NEWICK tree), allowing for specific subsequent data analysis or individual visualization steps by the user if desired.

**Post-processing of data.** Data post-processing starts with an initial removal step, where very rare OTUs can be dropped from analysis (*filter_otus_from_otu_table.py*, QIIME). OTUs can be omitted due to a low number of total reads on the one hand or due to rare occurrence within the samples on the other hand. To assess whether the sequencing effort (i.e. number of reads per sample) was sufficient, rarefaction curves are computed using Mothur's *rarefaction. single* command. The applied metric can be selected by the user out of the following possibilities: OTUs, Chao1 richness estimator, Shannon-Wiener diversity index, Simpson's diversity index and Good's coverage for OTUs. The output is provided either as text file containing all raw data or as line plot. Based on rarefaction curve analysis, the dataset can be subsampled to a unique read count for each sample to bring the data onto a common scale [61]. This step is done using the *single_rarefaction.py* script provided by the QIIME platform. The pseudo-random number generator used here for subsampling is an implementation of the Mersenne twister [62]. To facilitate use, CoMA provides the read count for each sample, sorted in increasing order. The next step allows for renaming sample names; all further results are presented using the newly assigned names. Finally, metadata can be assigned to the samples. These metadata can be used in all upcoming steps in order to group the samples.

**Data visualization and statistical appraisal.** First, summary files, containing the most abundant taxa of each sample at all taxonomic levels (kingdom, phylum, class, order, family, genus, and species) are created. The user can compute a report for bacteria, archaea, fungi and eukaryotes, as well as a general summary. Moreover, a summary report for a specific taxon can be created. During the next step, publication-ready and highly esthetic bar charts and heatmaps can be created for each taxonomic level. The user can choose between plots for bacteria, archaea, fungi, eukaryotes and prokaryotes. Alternatively, plots can be computed for a specific taxon, including all descendent taxa at the lower taxonomic levels. In addition, this step allows the user to set a threshold, based on relative abundance, for taxa to be included in the plots and a decision whether unassigned taxa will be included or excluded from the depictions. Moreover, the user can choose between various popular image file formats (EPS, JPEG, PDF, PNG, PS, RAW, SVG, TIFF) and adjust the pixel density if a raster graphic format was selected. Subsequently, Venn diagrams can be created to compare detected taxa among given groups for each taxonomic level. Within the next step, alpha diversity (or within-sample diversity) can be calculated using various different calculators (*observed OTUs*, *Shannon-Wiener index*, *Simpson's index*, *Pielou's evenness index*, *Good's coverage of counts*, *Chao1 richness estimator*, *Faith's phylogenetic diversity*). Every metric has different strengths and limitations—technical discussion of each metric is readily available online and in ecology textbooks. The output is provided either as raw-data text file or as publication-ready bar chart. Finally, CoMA offers two different approaches for analyzing beta diversity: ordination and hierarchical cluster analysis. Ordination can be done using various different metrics (*Minkowski*, *Euclidean*, *Manhattan*, *Cosine*, *Jaccard*, *Canberra*, *Chebyshev*, *Braycurtis*, *Dice*, *Weighted UniFrac*, *Unweighted UniFrac*). After calculating of a distance matrix, data are visualized in either two- or three-dimensional Principal coordinates analysis (PCoA) plots. To quantify the strength of the grouping, statistical tests (ANOSIM, PERMANOVA; 999 permutations) are applied. For cluster analysis, the user can choose between the following linkage methods for the cluster analysis: *Single*, *Complete*, *Average* (UPGMA), *Weighted* (WPGMA), *Centroid* (UPGMC), *Median* (WPGMC) and *Ward*. These methods are used to compute the distance d(s,t) between two clusters s and t, using a bottom-up approach. In addition, the user can select a suitable metric to determine the

distance between two individual data points: *Euclidean*, *Cosine*, *Cityblock*, *Correlation*, *Jaccard*, *Braycurtis* and *Dice*. Results from cluster analysis are presented as dendrogram plots.

## Mock community datasets

All mock community datasets for benchmarking CoMA were taken from the public *mockrobiota* repository [63]. In order to cover different input sequences in terms of variable region, sequence length and diversity, the three datasets *mock-13* [64], *mock-16* [65] and *mock-26* [66] were selected. *Mock-13* comprises 21 bacterial strains (18 genera; three replicates) that were sequenced on an Illumina MiSeq machine (paired-end run) by targeting the V4 region of the 16S rRNA. The same sequencing procedure was also used for *mock-16*; however, this dataset additionally includes archaea and the overall diversity is with 59 strains (46 genera; three replicates) much higher than in *mock-13*. *Mock-26* includes ITS data from 11 fungal strains (11 genera) that were sequenced with a *gx-flx-titanium* sequencer (Roche, Switzerland). The whole dataset comprises over 130 samples, including multiple replicates and different variations of the community (even abundance, uneven abundance, pure culture). For this work, samples Mock.81, Mock.92 and Mock.99 were randomly picked out of the evenly distributed samples.

**Mock data analysis with CoMA.** *Mock-13* forward and reverse reads were merged in a first step and the quality of the joined reads was then determined. Based on QC analysis, reads were selected due to their sequence length (240–265 bp) and their mean PQS ($> 15$). The reads were not trimmed since primers and other sequence appendices were already cut off in the raw input files. After the second QC step, reads were clustered into OTUs ($> 97\%$ similarity) and representative sequences were aligned to the SILVA database (SSU 132) using BLAST. The same procedure was applied for *mock-16*, however with two minor modifications. Before quality selection, primers were trimmed off from the forward (19 bp) and reverse end (20 bp). In addition, the PQS selection was more rigorous ($> 30$) since the overall quality of the reads was higher in *mock-16* compared with *mock-13*.

A slightly different protocol was applied for *mock-26* since the dataset was still multiplexed and contained single-end reads. Demultiplexing was done with the "demux emp-single" command within the QIIME2 toolkit since CoMA does not support multiplexed input files in its current version. Selected files (Mock.81, Mock.92, Mock.99) were then subjected to the CoMA pipeline, starting with the first QC step. Reads with a maximum length of 500 bp were then selected. A minimum sequence length was not adjusted since ITS reads typically show a wide distribution in sequence length and thus a loss of information should be avoided. Since the overall quality level of the sequences was high, a minimum average PQS $> 30$ was adjusted. After the second QC step and the OTU clustering, reads were aligned to the UNITE database (version 8) using the BLAST algorithm.

**Mock data analysis with Mothur.** Data analysis of *mock-13* and *mock-16* with Mothur (v. 1.39.5) was done according to the standard operating procedure (SOP) for Illumina MiSeq data, provided at the Mothur homepage (https://www.mothur.org/ wiki/MiSeq_SOP, [32]). Input forward and reverse reads were merged with "make.contigs". After checking the quality with "summary.seqs", ambiguous bases and reads with a length of $> 270$ bp (*mock-13*) and $> 292$ bp (*mock-16*) were removed using "screen.seqs". A table of unique sequences was created with "unique.seqs" and a prevailing counting table, providing the occurrences of the unique sequences in each sample, was created using "count.seqs". To prepare the SILVA (version 132) database, "pcr.seqs" was applied on the database file with the following limit settings: start = 13,862, end = 23,444. Unique sequences were aligned to the prepared database using the "align.seqs" tool and outliers were removed with "screen.seqs" (start = 8, end = 9,578 (*mock-13*) or 9,582 (*mock-16*), maxhomop = 8). After a sequence filtering step with "filter.

seqs" (vertical = T, trump =.), reads were again checked for unique sequences using the "unique.seqs" tool to exclude potential sequence redundancies introduced in course of the trimming step. For de-noising of data, sequences were pre-clustered with "pre.cluster" (diffs = 2) and chimeras were removed with "chimera.uchime" and a subsequent "remove. seqs" step. To get rid of sequencing errors, sequences were classified with "classify.seqs" and unknown reads (at kingdom level) as well as reads assigned to chloroplasts, mitochondria or eukaryota were removed with "remove.lineage". Sequences were then clustered using "cluster. split" (splitmethod = classify, taxlevel = 4, cutoff = 0.03) and a shared file was created with "make.shared". After taxonomic assignment ("classify.otu"), a BIOM file was created ("make. biom") and then transformed to a tab-delimited OTU-table using the independent "biom-convert" tool (http://biom-format.org/).

For *mock-26*, selected files (Mock.81, Mock.92, Mock.99) were imported using "make.file" and "fastq.info". Low-quality reads were then excluded ("trim.seqs"; qaverage = 15) and all individual files were merged to a combined FASTA file ("merge.files", "make.group"). After summarizing all data with "summary.seqs", reads with > 10 ambiguous bases were removed. A greater tolerance of ambiguous bases compared with *mock-13*/*mock-16* was required since all sequences constantly contained a considerable number of ambiguous bases. A sequence length selection was omitted due to the high inconsistency among the ITS data. After finding the unique sequences ("unique.seqs") and creating a count table ("count.seqs"), analysis was continued as described for the other mock communities, starting with the pre clustering step ("pre.cluster"). The newest Mothur-optimized UNITE release (version 7) was used as reference database for sequence alignment.

**Mock data analysis with QIIME.**   Data analysis of *mock-13* and *mock-16* with QIIME (v. 1.8.0) was done according to the recommendations for Illumina data on the QIIME homepage (http://qiime.org/tutorials/processing_illumina_data.html, [33]). First, paired-end input reads (all files in FASTQ format) were merged using the "multiple_join_paired_ends.py" script within the QIIME toolkit. This is an alternative version of the "join_paired_ends.py" script and allows the merging of already-demultiplexed FASTQ files. Quality filtering was done with the "multiple_split_libraries_fastq.py" script and chimeras were removed with "usearch61" using the "identify_chimeric_seqs.py" and "filter_fasta.py" scripts. Reads were then clustered into OTUs with the "usearch61" method of "pick_otus.py" and representative (= most abundant) sequences for each OTU were picked using "pick_rep_set.py". Representatives were then assigned to the QIIME-optimized Silva SSU database (release 132) with the "assign_taxonomy. py" script (assignment method: uclust) and the OTU-table was constructed using "make_otu_-table.py". The created BIOM file was converted to a tab-delimited OTU-table with the external "biom-convert" tool as described in the previous chapter.

Selected input files for *mock-26* (Mock.81, Mock.92, Mock.99) underwent, in a first step, a quality selection ("split_libraries_fastq.py"), where reads with an average PQS < 15 were removed. All upcoming steps were done as described above for the other mock communities. The newest QIIME-optimized UNITE release (version 8, dynamic construction) served as reference database for sequence alignment ("assign_taxonomy.py").

**Mock data analysis with QIIME2.**   Data analysis with QIIME2 (v. 2019.10) followed the "Moving pictures" tutorial, accessible on the program's webpage (https://docs.qiime2.org/2019.10/tutorials/moving-pictures/). However, *mock-13* and *mock-16* required a slightly alternative procedure due to their paired-end data structure, and the files from all mock communities (*mock-13*, *mock-16*, *mock-26*) needed to be imported in a different manner since the data were already demultiplexed.

First, *mock-13* and *mock-16* input files were imported using "qiime tools import" with the settings "*SampleData[PairedEndSequencesWithQuality]*" as data type and

"*CasavaOneEightSingleLanePer- SampleDirFmt*" as input format. After summarizing and quality-checking the imported data ("qiime demux summarize"), "qiime dada2 denoise-paired" was applied. This central step calls the DADA2 pipeline and provides several functions, including sequence trimming and truncation, quality filtering, removal of phiX reads, chimera removal, and the construction of a feature table (suitable filtering settings resulted from the previous quality-checking step). In contrast to all other pipelines used for this study, DADA2 provides ASVs, which are created by grouping unique sequences. ASVs are equivalent to OTUs at a 100% cutoff level and are discussed to replace customary OTUs (97% cutoff) in the future. Therefore, QIIME2 always means QIIME2-DADA2 within the context of this publication since the pipeline also offers other options for sequence clustering (e.g. Deblur) and results may considerably differ. For *mock-16*, 19 and 20 bp were trimmed off from the forward and reverse end, respectively. No trimming was applied for *mock-13* since all sequence appendices were already removed in the raw input files. Thereafter, taxonomic information was assigned to the representative sequences using a pre-trained Naive Bayes classifier (Silva 132, 99% OTUs from the 515F/806R inner primer region, available at https://docs.qiime2.org/2019.10/data-resources/) and the "qiime feature-classifier classify-sklearn" tool. After exporting the table- and taxonomy QIIME2 artifact with "qiime tools export" and the external "biom-convert" tool, a tab-delimited ASV-table file was created.

For *mock-26*, selected input files (Mock.81, Mock.92, Mock.99) were imported with "qiime tools import" (data type: "*SampleData[SequencesWithQuality]*", input format: "*CasavaOneEightSingle-LanePerSampleDirFmt*"). Sequence preparation and ASV-table construction was again done with the DADA2 pipeline; however, in this case with the single-end version of the plugin ("qiime dada2 denoise-single"). No trimming or sequence truncation was applied. Representative sequences were then taxonomically assigned with a pre-trained UNITE classifier and the "qiime feature-classifier classify-sklearn" plugin. The classifier was trained on the newest UNITE database release (version 8, dynamic construction), and the reads were not trimmed to the ITS primer sites as suggested by the QIIME2 documentation. The final tab-delimited ASV-table file was created as previously described for *mock-13* and *mock-16*.

## Soil dataset

The chosen dataset for demonstration of the CoMA pipeline originated from a field experiment conducted at the Department of Microbiology, University of Innsbruck, in autumn 2016. It aimed at investigating the microbial community structure in soils from three different habitats: forest (F), grassland (GR) and swamp (S), characterized by different nutrient contents, a specific vegetation and different degrees of human influence. However, all sites were located in close proximity and thus subjected to similar geological and climatic conditions.

**Soil sampling.** Samples were taken in Trins, Tyrol (Austria; 47˚04′59" N, 11˚25′00" E) on November 28, 2016. The annual average temperature is 4˚C and the annual precipitation 770 mm [67]. Samples were taken at the end of the vegetation period, two months after the last fertilization with manure (swamp samples). Details on the sampling sites are given in Table 2.

From each site, four field replicates were taken within an area of approximately 100 $m^2$. Each field replicate was composed of nine cores (0–10 cm depth, Ø 2 cm) taken in a regular systematic grid strategy (within a square of 1.4 x 1.4 m). Immediately after the sampling, soil samples were sieved (< 2 mm) and stored at -20˚C until use. Soil physico-chemical characterization (S1 Table) was done as described by Fernández-Delgado Juárez et al. [69].

**DNA extraction and NGS.** Total DNA was extracted using the NucleoSpin Soil kit (Macherey-Nagel, Düren, Germany) according to the manufacturer's protocol with minor adaptations: 0.25 g (fresh weight) of twelve soil samples (stored at -20˚C) were lysed in Lysis

Table 2. Sites that were selected for soil analysis in order to demonstrate the functionality of CoMA.

| Habitat | GPS-coordinates (WGS 84) | Characteristics |
|---------|--------------------------|-----------------|
| F | 47˚04'55" N, 11˚25'47" E | Conifer forest, mainly *Picea abies*; Southern slope; soil type: medium-developed sediment brown earth, Leptic Cambisol. |
| GR | 47˚04'57" N, 11˚25'13" E | Meadow; not fertilized for > 10 years; cut twice a year; soil type: loose sediment brown earth, Eutric Cambisol. |
| S | 47˚04'27" N, 11˚24'22" E | Former swamp, drained in the 1980's; water level between 5–50 cm, depending on the season; manure amendment: 2–3 times a year; cut twice a year; soil type: calcareous extreme gley, Stagnosol. |

All areas were located in Gschnitz valley, Tyrol (Austria) near the village of Trins. Information on the soil type was gained from the Austrian Soils Map of the Austrian Research Centre for Forests (https://bodenkarte.at/) and the world reference base for soil resources [68]. F = forest. GR = grassland. S = swamp.

Buffer 1 (SL1) at room temperature using a horizontal shaker (MM 2000, Retsch, Haan, Germany) at an amplitude of 80% for 5 min. Elution was done in 2x40 µL of Buffer SE. DNA extracts were checked on a 1.5% agarose gel and DNA concentration was determined via QuantiFluor dsDNA dye measurement using the Quantus fluorometer (Promega, Mannheim, Germany). Extracts were stored in low-DNA binding tubes (Axygen, Corning, USA) at -20˚C until sequencing. Samples were then subjected to NGS amplicon-sequencing (Microsynth AG, Balgach, Switzerland) on an Illumina MiSeq device using a 250 bp paired-end (v2) approach targeting the V4 region of the 16S SSU rRNA gene. The applied primer pair was 515f (5'-GTGCCAGCMGCCGCG GTAA-3') and 806r (5'-GGACTACHVGGGTWTCTAAT-3'), as described by Caporaso et al. [70] and recommended for soil in the Earth Microbiome Project [71].

**Soil data analysis with CoMA.** Data analysis with CoMA (quality selection, OTU clustering, taxonomic assignment) was done as previously described for *mock-13* but with a PQS cutoff $\geq$ 30. Rare OTUs with a sum of reads < 50 within all samples were excluded (corresponding ~0.005% of all reads as previously described in other studies [72, 73]) and data was subsampled to a depth of 92,539 after checking rarefaction curves (S1 Fig). This corresponds to the lowest read count among all samples and all analysis pipelines (CoMA, Mothur, QIIME). At this depth, the rarefaction curves of all samples reached a steady state and thus the sequencing effort was assumed to suffice. Diversity analysis was done using the four provided diversity indices: number of OTUs, Shannon-Wiener, Chao1 and Simpson. For cluster analysis, UPGMA (CoMA input: "average") was chosen as method and Braycurtis distance as metric. This combination is suitable for grouping microbial communities and therefore often used in microbial ecology studies [74].

**Soil data analysis with Mothur.** Soil samples were analyzed with Mothur following the same protocol and settings as described for *mock-13*. Removal of rare OTUs (sum of reads < 50) as well as data subsampling (92,539 reads) was done with CoMA as described above.

**Soil data analysis with QIIME.** Soil data analysis was done following the same protocol and settings as described previously for *mock-13* and *mock-16*. OTUs removal and subsampling was again done with CoMA using the same settings.

## Statistical analyses

Principal component analysis (PCA) was done with the Canoco 5 software package for multivariate data analysis [75]. Microbial families, excluding unassigned taxa, were used as dataset. Prior to analysis, data were centered and log-transformed. The depiction of Canoco output

files was thereafter performed with python. Descriptive statistics, analysis of variance (ANOVA) and post-hoc tests were done using SPSS Statistics 24 (IBM, New York, USA) and Excel 2016 (Microsoft, Washington, USA). Post-hoc tests were calculated using the Bonferroni correction. Venn diagrams and cluster analysis were created with in-house developed Python scripts. Dendrograms were created with the UPGMA method as bottom-up approach. The data points were compared either with the Braycurtis distance (to compare different soil microbiomes) or with the Cosine distance (to compare different analysis platforms). Key microbial families were defined as those including > 5% of all assigned reads within each sample. The normality of the data was tested prior to analysis and the significance level was determined at p ≤ 5%.

## Results

### Benchmarking with mock communities

*Mock-13* comprised 21 bacterial strains that can be classified to 18 different genera: *Acinetobacter*, *Actinomyces*, *Bacillus*, *Bacteroides*, *Clostridium*, *Deinococcus*, *Enterococcus*, *Escherichia*, *Helicobacter*, *Lactobacillus*, *Listeria*, *Neisseria*, *Porphyromonas*, *Propionibacterium*, *Pseudomonas*, *Rhodobacter*, *Staphylococcus* (two strains) and *Streptococcus* (three strains). All strains were equally distributed and thus each genus corresponded to 4.76% of the total abundance, except for *Staphylococcus* (9.52%) and *Streptococcus* (14.29%). Data analysis with CoMA revealed 16 out of 18 genera, similar to Mothur and QIIME (Table 3). QIIME2 was able to find all genera except for *Propionibacterium*, which was not found with any of the pipelines (Fig 2). This, however, is a known issue and several other studies were not able to detect *Propionibacterium* in course of mock community tests [76]. The reason for that may be mismatches close to the 3' terminus of the 16S rRNA primer, leading to a low abundance or even a complete missing out of this genus [77]. Apart from *Propionibacterium*, *Actinomyces* (CoMA), *Bacillus* (Mothur) and *Clostridium* (QIIME) were not consistently detected. When looking at the average deviation per taxon, all pipelines performed similarly well. With a mean deviation of 3.11%, Mothur was most precise, closely followed by CoMA (3.27%) and the two QIIME versions (QIIME: 3.50%, QIIME2: 3.45%). On the other hand, Mothur yielded the highest proportion of erroneously assigned sequences (11.85%), followed by QIIME (9.20%). Both, CoMA and QIIME2 showed error rates below 1% (0.64% and 0.83%, respectively). Cluster analysis revealed smallest cosine distance between CoMA and QIIME2 (0.06; S2 Fig). Mothur and QIIME formed a second cluster, showing a distance of 0.15. All pipelines were closer related to each other than to the set point with an overall distance of 0.22.

The second 16S mock community included, beside bacterial, also archaeal strains. In total, *mock-16* comprised 59 strains that can be assigned to 46 different genera (archaea: 8, bacteria: 38). All strains were equally distributed, resulting in the following abundances at genus level:

**Table 3. Results of the benchmark test for four different analysis platforms.**

| Dataset | Detected Genera | | | | Average deviation per genus [%] | | | |
|---|---|---|---|---|---|---|---|---|
| | CoMA | Mothur | QIIME | QIIME2 | CoMA | Mothur | QIIME | QIIME2 |
| *mock-13* | 16/18 | 16/18 | 16/18 | 17/18 | 3.27 | 3.11 | 3.50 | 3.45 |
| *mock-16* | 43/46 | 40/46 | 44/46 | 34/46 | 1.22 | 1.52 | 1.22 | 2.09 |
| *mock-26* | 11/11 | 11/11 | 11/11 | 11/11 | 2.31 | 2.09 | 4.01 | 2.05 |
| *mock-13/mock-16/mock-26* | 70/75 | 67/75 | 71/75 | 62/75 | 1.89 | 1.99 | 2.17 | 2.41 |

Numbers of detected genera and accuracy, given as average deviation per genus, are shown (*mock-13*: 18 bacterial genera (16S rRNA); *mock-16*: 46 archaeal/bacterial genera (16S rRNA); *mock-26*: 11 fungal genera (ITS)).

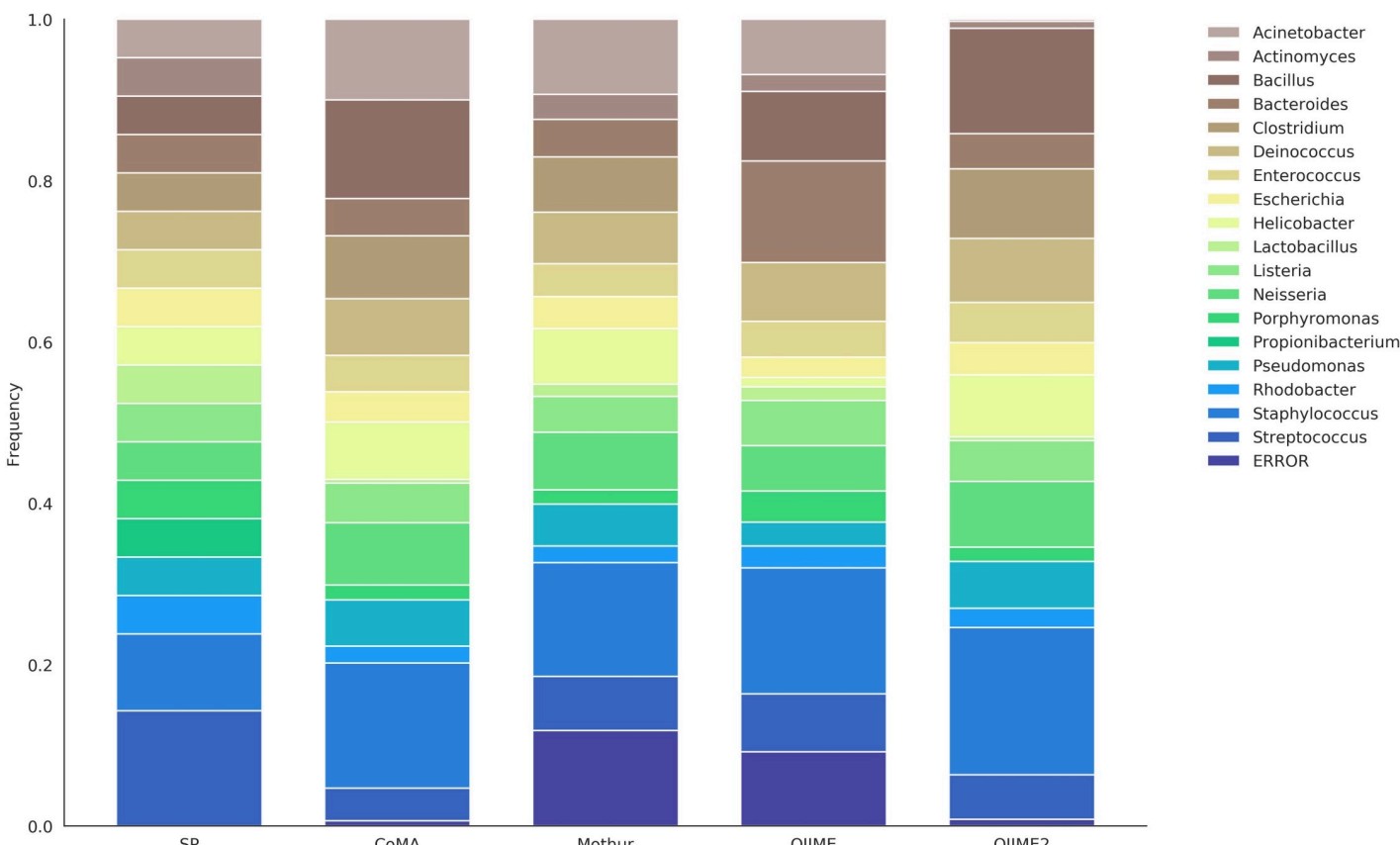

**Fig 2. Community composition of the *mock-13* dataset, revealed with four different analysis platforms.** The set point (SP) depicts the theoretically expected distribution and serves as reference. The dataset comprised 18 bacterial genera, targeted with 16S rRNA amplicon sequencing.

*Acidobacterium* (1.69%), *Akkermansia* (1.69%), *Anaerocellum* (1.69%), *Archaeoglobus* (1.69%), *Bacteroides* (3.39%), *Bordetella* (1.69%), *Burkholderia* (1.69%), *Caldicellulosiruptor* (1.69%), *Chlorobaculum* (1.69%), *Chlorobium* (5.08%), *Chloroflexus* (1.69%), *Clostridium* (1.69%), *Deinococcus* (1.69%), *Desulfovibrio* (3.39%), *Dictyoglomus* (1.69%), *Enterococcus* (1.69%), *Erwinia* (1.69%), *Fusobacterium* (1.69%), *Gemmatimonas* (1.69%), *Herpetosiphon* (1.69%), *Hydrogenobaculum* (1.69%), *Ignicoccus* (1.69%), *Leptothrix* (1.69%), *Methanocaldococcus* (1.69%), *Methanococcus* (3.39%), *Nanoarchaeum* (1.69%), *Nitrosomonas* (1.69%), *Nostoc* (1.69%), *Pelodictyon* (1.69%), *Persephonella* (1.69%), *Porphyromonas* (1.69%), *Pyrobaculum* (3.39%), *Pyrococcus* (1.69%), *Rhodopirellula* (1.69%), *Rhodospirillum* (1.69%), *Ruegeria* (1.69%), *Salinispora* (3.39%), *Shewanella* (3.39%), *Sulfitobacter* (3.39%), *Sulfolobus* (1.69%), *Sulfurihydrogenibium* (3.39%), *Thermoanaerobacter* (1.69%), *Thermotoga* (5.08%), *Thermus* (1.69%), *Treponema* (3.39%) and *Zymomonas* (1.69%). With CoMA, 43 out of 46 genera were correctly assigned (Table 3). QIIME detected 44, whereas Mothur and QIIME2 found 40 and 34, respectively. None of the pipelines was able to reveal *Rhodospirillum*, while *Clostridium* (Mothur) and *Ruegeria* (QIIME) were only found with one of the four applied analysis tools (Fig 3). All other genera were at least detected by two pipelines, 32 of them by all of them. CoMA and QIIME turned out to be most precise for this dataset, resulting in an average deviation of 1.22% per taxon. Accuracy rates of Mothur and QIIME2 were 1.52% and 2.09%, respectively. Comparing the proportion of wrongly assigned or unassigned genera, CoMA was most efficient (3.83%), followed by QIIME (3.88%). Mothur and QIIME2 both revealed more than one erroneous

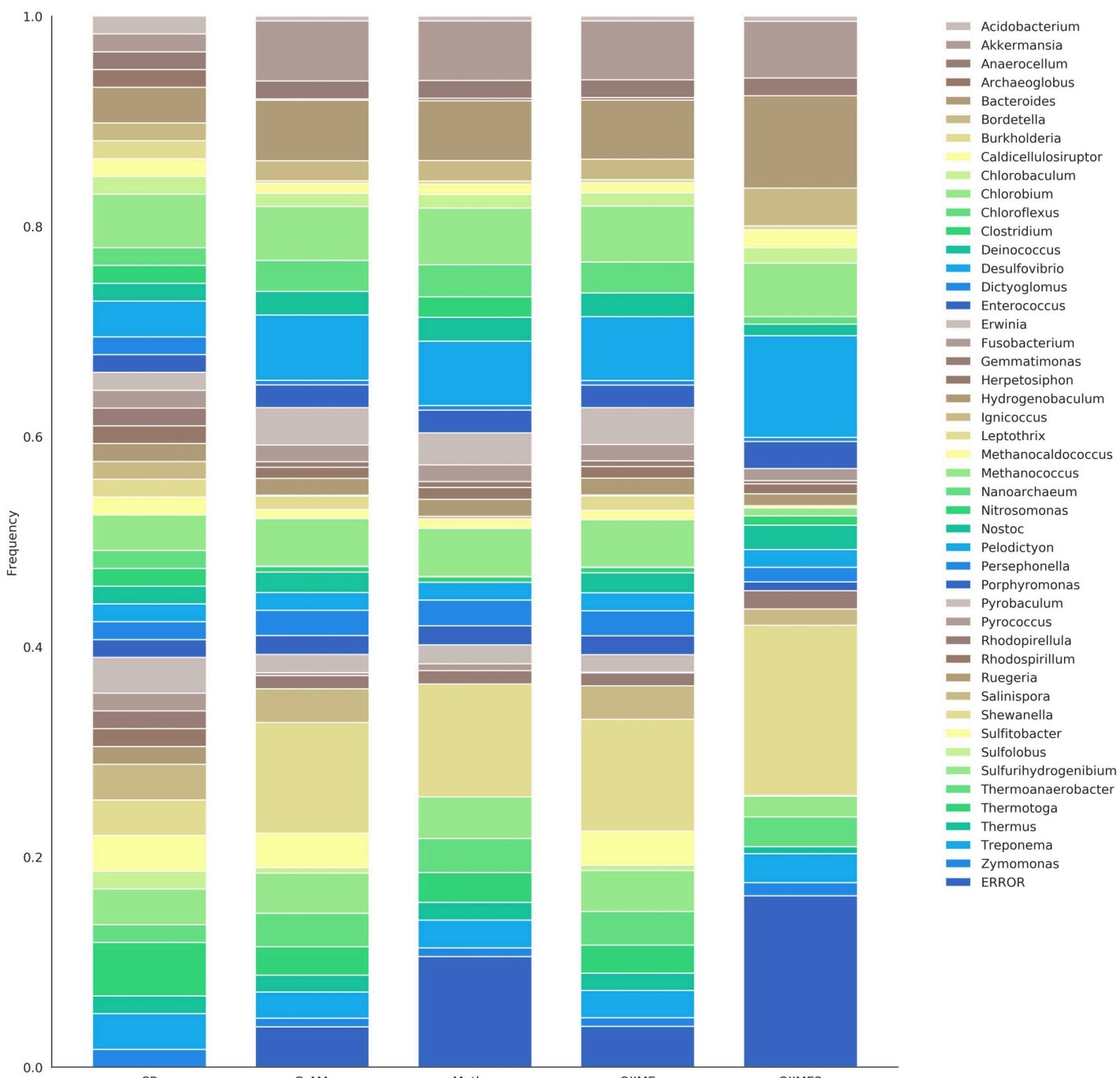

**Fig 3. Community composition of the *mock-16* dataset, revealed with four different analysis platforms.** The set point (SP) depicts the theoretically expected distribution and serves as reference. The dataset comprised 46 archaeal and bacterial genera, targeted with 16S rRNA amplicon sequencing.

genus out of ten tested sequences (10.53% and 16.31%, respectively). Cluster analysis based on the *mock-16* results revealed a cosine distance < 0.01 between CoMA and QIIME, indicating a very high similarity between these two pipelines (S3 Fig). With a cosine distance of 0.08, Mothur was closer related to them than to QIIME2. However, as seen before for *mock-13*, all applied pipelines were much closer related to each other (maximum distance: 0.15) than to the set point (overall distance: 0.30).

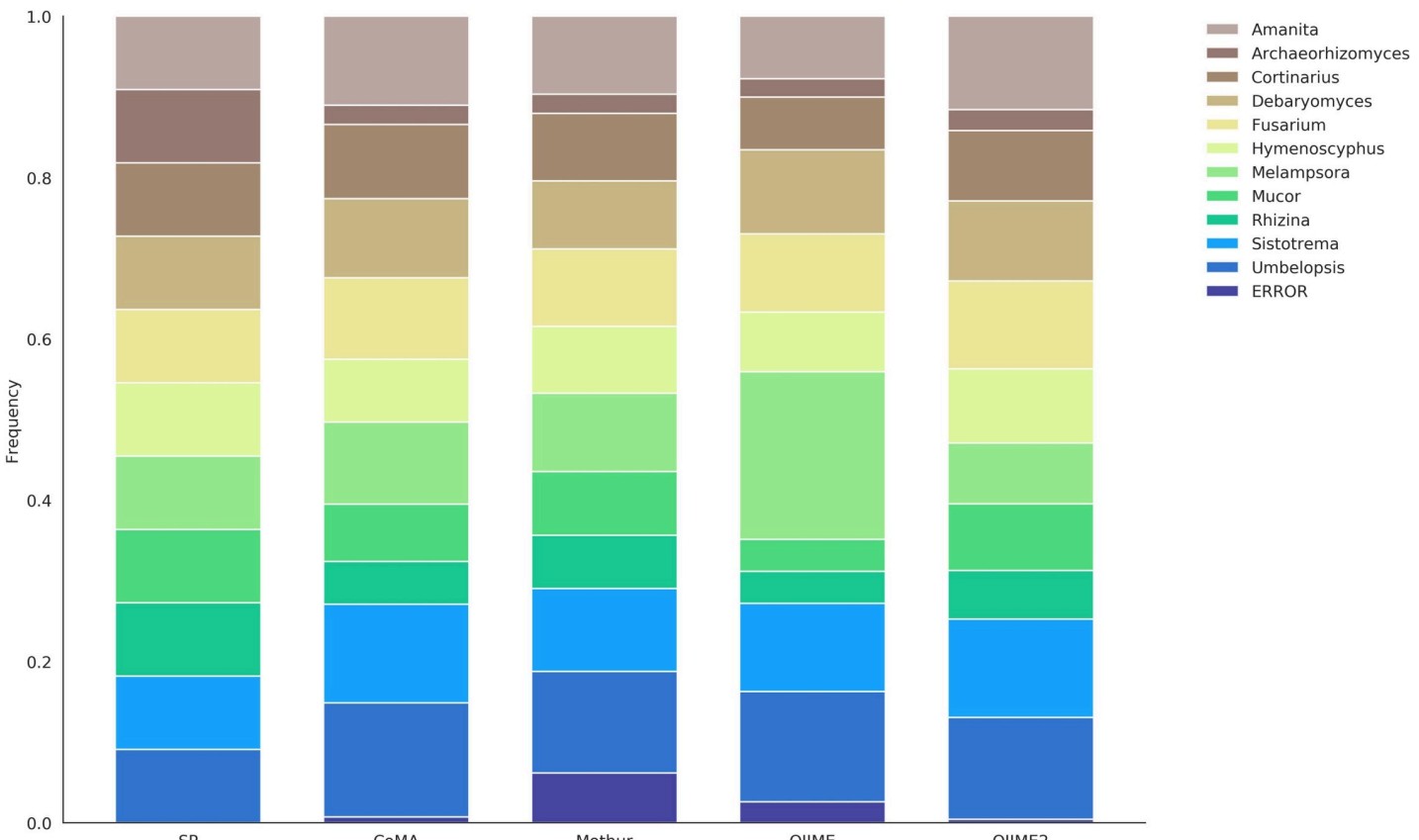

**Fig 4. Community composition of the *mock-26* dataset, revealed with four different analysis platforms.** The set point (SP) depicts the theoretically expected distribution and serves as reference. The dataset comprised 11 fungal genera, targeted with ITS amplicon sequencing.

The ITS dataset (*mock-26*) included 11 fungal strains, assigned to 11 different genera: *Amanita*, *Archaeorhizomyces*, *Cortinarius*, *Debaryomyces*, *Fusarium*, *Hymenoscyphus*, *Melampsora*, *Mucor*, *Rhizina*, *Sistotrema*, *Umbelopsis*. All taxa were equally distributed, resulting in an abundance of 9.09% per genus. Contrary to *mock-13* and *mock-16*, all genera of this dataset were detected with each of the four pipelines (Table 3, Fig 4). QIIME2 and Mothur showed the highest accuracy with an average deviation of 2.05% and 2.09% per taxon, respectively, followed by CoMA (2.31%) and QIIME (4.01%). QIIME2 was also superior in terms of wrongly assigned or unassigned sequences at genus level for this dataset (0.44%), slightly lower than CoMA with an error rate of 0.71%. QIIME and Mothur followed with distinctively higher error rates of 2.62% and 6.17%, respectively. Cluster analysis revealed smallest cosine distance between the results computed with CoMA and QIIME2 ($< 0.01$; S4 Fig). Mothur was slightly more distant (0.02) but still closer related than to the set point with an overall distance of 0.05 to the three aforementioned pipelines. For the *mock-26* dataset, QIIME was far distant to the set point as well as to the other analysis tools (0.10).

Taking all mock communities together, CoMA showed the overall highest accuracy with an average deviation of 1.89% per taxon (Table 3), compared to Mothur (1.99%), QIIME (2.17%) and QIIME2 (2.41%). Looking at the hierarchical cluster analysis, CoMA and QIIME2 showed the smallest cosine distance (0.08; S5 Fig). Mothur followed with a distance of 0.11 and QIIME with a distance of 0.12. All four pipelines were much closer related to each other than to the estimated values (overall cosine distance: 0.18).

## Soil dataset as practical example

In addition to mock communities, also real soil samples were analyzed with CoMA, Mothur and QIIME. A detailed description of the microbiological characteristics of the investigated soil samples determined with CoMA can be found in the supplementary material (S1 File). In the main article, we are focusing on the comparison between the three different analysis tools.

Data analyses with Mothur and QIIME resulted in the same pattern of Shannon-Wiener diversity (H', Fig 5), Chao1 diversity, Simpson diversity (D) and OTU richness (S6 Fig) as seen for the CoMA pipeline (S1 File). However, the level differed between the applied analysis tools. Mothur generally evoked higher diversity (H': p = 0.002, Chao: p < 0.001, D: p = 0.009) and OTU richness (p = 0.004) than CoMA, irrespective of the diversity calculator and the habitat. QIIME data were statistically neither distinguishable to CoMA nor to Mothur, except for the Chao index where QIIME resulted in a higher diversity than CoMA (p = 0.029). When looking at each habitat individually, Shannon-Wiener diversity reached the highest value with Mothur, followed by QIIME and CoMA in all cases.

Principal component analysis based on microbial families clustered the CoMA and QIIME samples in close proximity to each other (Fig 6). Data points from grassland and swamp were located in the third quadrant, showing minor spatial deviation, mainly along the PCA Axis 2. The first two PCA axes explained 74% of the total variation within the taxonomic data (53% and 21% for PCA axis 1 and 2, respectively). Forest samples showed a greater distance to the two other habitats and were located in the second quadrant. Samples analyzed with Mothur were located in the positive direction of PCA Axis 1 and thus clearly separated from the data points of CoMA and QIIME. Data points representing the three habitats showed, however, a similar pattern along PCA Axis 2 as those obtained by the other pipelines.

To dig deeper into the differences/similarities between the three applied analysis pipelines, numbers of detected taxa were determined for each taxonomic level (Fig 7). Data analysis with CoMA resulted in the highest number of classes and families (114 and 281, respectively) and QIIME of phyla and genera (45 and 415, respectively). Mothur analysis always resulted in the lowest numbers of taxa, irrespective of the taxonomic level. The analysis showed that CoMA and QIIME detected a similar taxa composition at each level. Mothur, however, did not reveal common taxa at a comparable rate but detected several taxa that were neither found with CoMA nor with QIIME. At phylum level, 39 out of 54 taxa (72%) were found with both CoMA and QIIME. Out of these 39 common taxa, 19 were also detected with Mothur (35%).

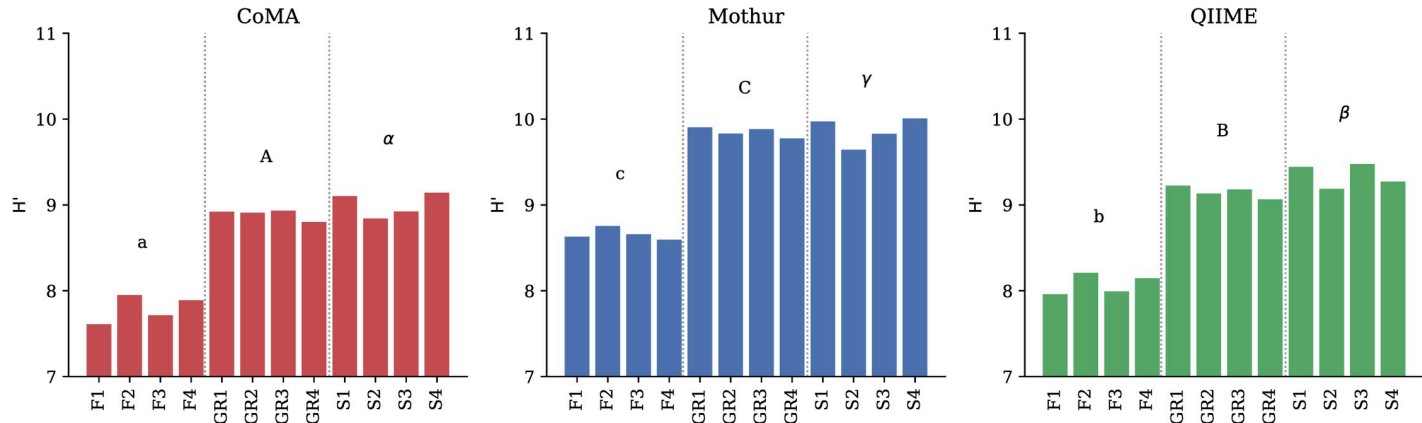

**Fig 5. Shannon-Wiener diversity (H') of the three different soils after sequencing data analysis with CoMA, Mothur and QIIME.** Four replicates for each habitat are shown. Letters indicate significant differences across the analysis tools for each habitat. F = forest. GR = grassland. S = swamp.

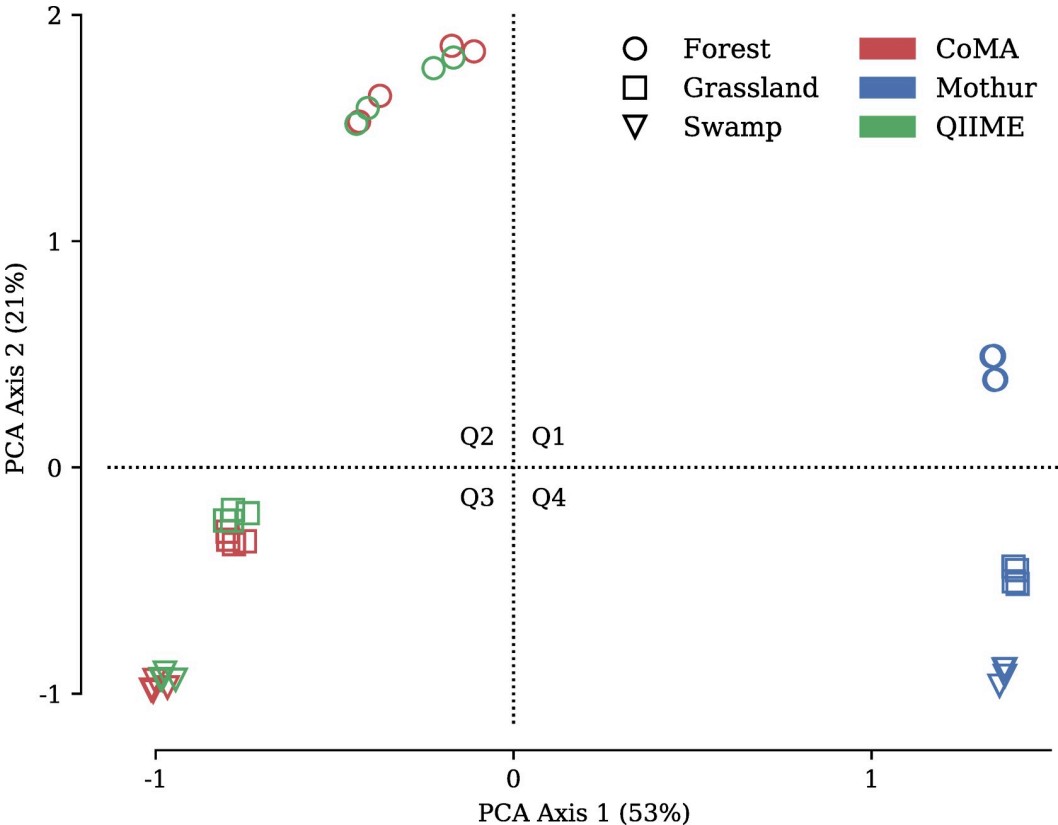

**Fig 6. Principal component analysis based on archaeal and bacterial families of soil samples from three different habitats: Forest, grassland and swamp.** The color code indicates the applied data analysis tool: CoMA, Mothur and QIIME. Q1—Q4 = quadrants of the coordinate system.

CoMA and QIIME shared 100 (61%) and 98 (60%) out of all detected 164 classes, respectively, while Mothur found only 27 shared classes (16%) with the other pipelines. The same trend was observed at order level where CoMA and QIIME detected 183 (68%) and Mothur 50 (19%) out of 270 taxa, and at family level where CoMA found 243 (70%), QIIME 245 (70%) and Mothur 88 (25%) out of 349 families. However, at genus level, the percentage of taxa detected with all three of the analysis tools was much higher compared to the other taxonomic levels, except for phylum level. CoMA revealed 317 (61%), QIIME 339 (65%) and Mothur 159 (31%) out of 518 genera. The percentage of taxon overlap for the three analysis tools was higher at phylum level (35% of all phyla), followed by genus (26%) and family level (24%). At order and class levels, the percentages of shared taxa were much lower and accounted for only 18% and 15%, respectively. CoMA and QIIME overlapped in 83%, 78%, 82%, 80% and 72% of all commonly found taxa at phylum, class, order, family and genus level, respectively.

To compare the general output of the pipelines and evaluate CoMA as an alternative analysis tool, key families (i.e. abundance > 5% of reads) of the three habitats were analyzed. Within the forest samples, all families determined with QIIME were also found with CoMA (11 out of 16, Table 4). Eight out of these eleven families were also detected with Mothur; however, Chthoniobacteraceae, Nitrosomonadaceae and Solibacteraceae were not found with Mothur. On the other hand, Mothur resulted in five unique families (Bradyrhizobiaceae, Comamonadaceae, Nitrososphaeraceae, Oxalobacteraceae, Planctomycetaceae) that were not caught with either of the other two tools. In grassland, nine out of 18 key families were identified with all

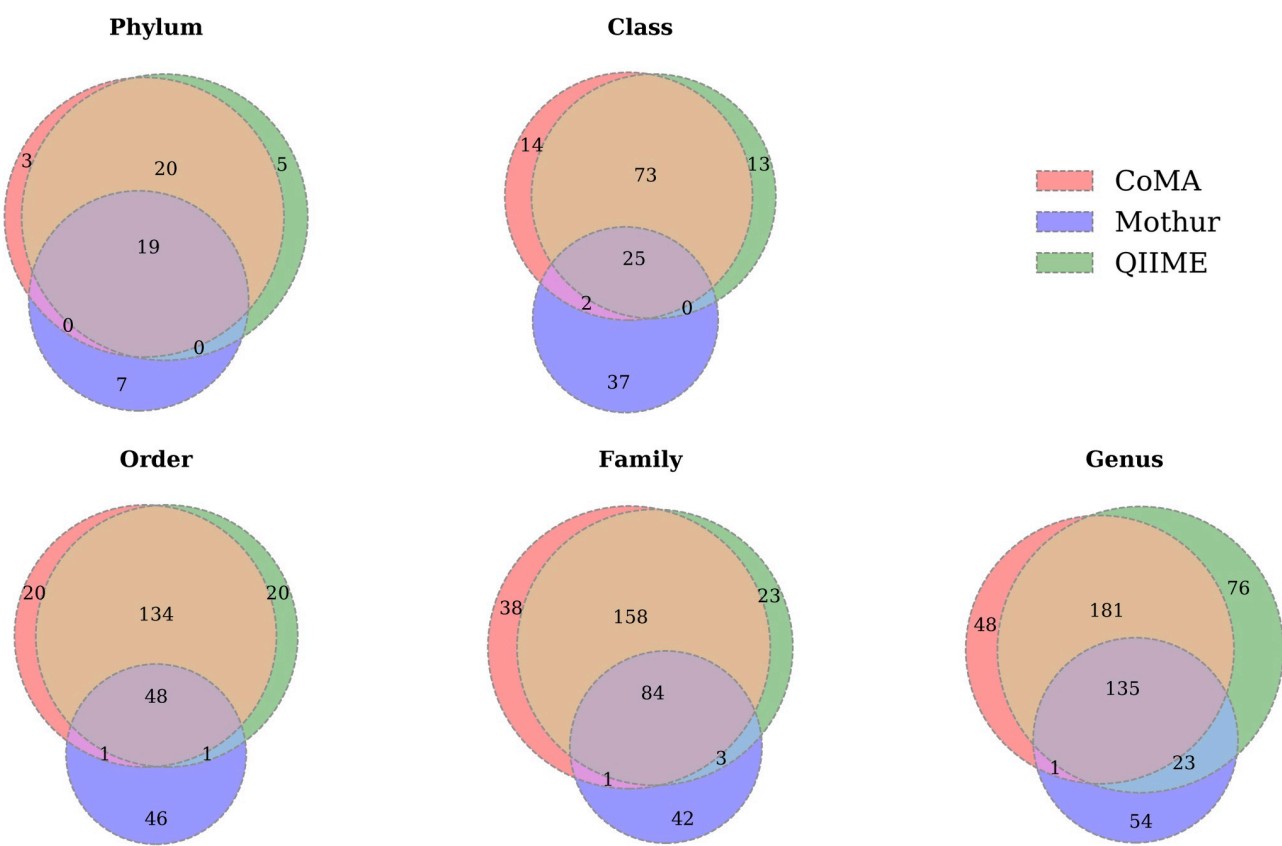

**Fig 7. Venn plots showing the shared phyla, classes, orders, families and genera found with CoMA, Mothur and QIIME in the soil samples.** Data include all of the three investigated habitats (forest, grassland, swamp).

pipelines. CoMA and QIIME determined again the same taxa (14 out of 18), but did not expose Bradyrhizobiaceae, Comamonadaceae, Oxalobacteraceae and Planctomycetaceae, which were found with Mothur. Mothur, in return, did not detect Burkholderiaceae, Chthoniobacteraceae, Nitrosomonadaceae, Solibacteraceae and Xanthobacteraceae in the analyzed soil samples. The same occurred in the swamp soil. All key families detected by QIIME were also identified with CoMA (15 out of 19). Ten of them were equally found with Mothur, except for Burkholderiaceae, Chthoniobacteraceae, Nitrosomonadaceae, Solibacteraceae and vadinHA17 (order: Bacteroidales). On the other hand, Mothur determined four families (Bradyrhizobiaceae, Comamonadaceae, Oxalobacteraceae, Planctomycetaceae) that were neither found with CoMA nor with QIIME. Six key families were identified with all of the three analysis tools irrespective of the soil type: Anaerolineaceae, Chitinophagaceae, Flavobacteriaceae, Gaiellaceae, Mycobacteriaceae and Sphingobacteriaceae.

Irrespective of the soil type, CoMA and QIIME identified more taxa than Mothur (p = 0.024, p = 0.032), except for the kingdom and species level, where all of the three pipelines classified 100% and 0–2%, respectively (Table 5). At phylum level, CoMA and QIIME assigned all reads to known taxa whereas Mothur failed in 7–13%. Generally, the classification percentage decreased with lower taxonomic levels (p < 0.001). CoMA and QIIME behaved similarly at all taxonomic levels; however, CoMA identified slightly more orders and families when grassland was analyzed (+5% and +3%, respectively; p < 0.001).

**Table 4. Key microbial families found in three different habitats: Forest, grassland and soil.**

| Family | Forest | | | Grassland | | | Swamp | | |
|---|---|---|---|---|---|---|---|---|---|
| > 5% | CoMA | Mothur | QIIME | CoMA | Mothur | QIIME | CoMA | Mothur | QIIME |
| Anaerolineaceae | 136 ± 40 | 53 ± 13 | 152 ± 35 | 245 ± 18 | 118 ± 10 | 241 ± 26 | 2,754 ± 678 | 2,513 ± 580 | 2,669 ± 643 |
| Bradyrhizobiaceae | | 4,302 ± 575 | | | 744 ± 63 | | | 607 ± 134 | |
| Burkholderiaceae | 1,620 ± 639 | 869 ± 248 | 1,642 ± 665 | 1,836 ± 109 | | 1,764 ± 159 | 5,685 ± 757 | | 5,939 ± 900 |
| Chitinophagaceae | 2,079 ± 746 | 2,443 ± 988 | 2,400 ± 979 | 1,565 ± 266 | 1,277 ± 210 | 1,265 ± 213 | 1,693 ± 212 | 1,500 ± 171 | 1,437 ± 163 |
| Chthoniobacteraceae | 9,001 ± 3,386 | | 7,983 ± 2,928 | 926 ± 64 | | 847 ± 43 | 374 ± 73 | | 356 ± 66 |
| Comamonadaceae | | 178 ± 150 | | | 461 ± 35 | | | 2,311 ± 674 | |
| Flavobacteriaceae | 349 ± 297 | 282 ± 243 | 278 ± 239 | 848 ± 178 | 784 ± 237 | 732 ± 183 | 9,513 ± 1,400 | 8,816 ± 1,314 | 7,939 ± 1,259 |
| Gaiellaceae | 102 ± 75 | 1,530 ± 1,232 | 113 ± 106 | 2,619 ± 352 | 7,127 ± 766 | 2,951 ± 370 | 430 ± 92 | 641 ± 150 | 490 ± 90 |
| Geobacteraceae | | | | 89 ± 33 | 69 ± 26 | 72 ± 29 | 2,536 ± 251 | 2,247 ± 206 | 2,153 ± 199 |
| Mycobacteriaceae | 1,792 ± 772 | 1,615 ± 714 | 1,488 ± 647 | 2,023 ± 245 | 1,779 ± 205 | 1,671 ± 183 | 192 ± 27 | 172 ± 23 | 163 ± 15 |
| Nitrosomonadaceae | 514 ± 468 | | 640 ± 523 | 2,262 ± 194 | | 2,433 ± 215 | 2,656 ± 365 | | 4,386 ± 356 |
| Nitrososphaeraceae | | 253 ± 305 | | 2,785 ± 411 | 2,613 ± 422 | 2,372 ± 342 | 76 ± 20 | 74 ± 18 | 63 ± 11 |
| Oxalobacteraceae | | 229 ± 99 | | | 210 ± 12 | | | 2,081 ± 563 | |
| Planctomycetaceae | | 3,650 ± 1,045 | | | 1,902 ± 125 | | | 1,392 ± 112 | |
| Propionibacteriaceae | | | | 2,813 ± 268 | 2,522 ± 234 | 2,815 ± 259 | 69 ± 20 | 62 ± 21 | 78 ± 19 |
| Solibacteraceae | 3,790 ± 994 | | 3,929 ± 1,040 | 311 ± 19 | | 326 ± 36 | 709 ± 81 | | 719 ± 43 |
| Sphingobacteriaceae | 1,861 ± 847 | 1,471 ± 674 | 1,470 ± 697 | 90 ± 29 | 62 ± 19 | 67 ± 19 | 521 ± 94 | 436 ± 94 | 418 ± 75 |
| vadinHA17 | | | | | | | 3,407 ± 1,095 | | 2,779 ± 892 |
| Xanthobacteraceae | 7,373 ± 564 | 111 ± 77 | 8,374 ± 749 | 2,090 ± 138 | | 2,228 ± 147 | 1,663 ± 151 | 136 ± 7 | 1,888 ± 80 |

Data (means ± standard deviation (n = 4)) were determined with three different analyses tools: CoMA, Mothur and QIIME. Families with a relative abundance < 5% and unassigned taxa were excluded from the table, and families with a read count < 50 were considered as absent. Differently colored cells indicate the three most abundant families.

## Discussion

### Validation of CoMA

The main purpose of this work was to validate CoMA by comparing the results with those of already established tools. Mothur and QIIME, the two currently most popular analysis tools for amplicon-sequencing data were chosen as reference for both test assays: a benchmark test using three different mock communities, covering different variable regions, sequence lengths and abundances, as well as the analysis of a real soil dataset. For the benchmark test, QIIME2

**Table 5. Unclassified reads for each taxonomic level in three different soils: Forest, grassland and swamp.**

| Unclassified [%] | Forest | | | Grassland | | | Swamp | | |
|---|---|---|---|---|---|---|---|---|---|
| | CoMA | Mothur | QIIME | CoMA | Mothur | QIIME | CoMA | Mothur | QIIME |
| Kingdom | 0 ± 0 | 0 ± 0 | 0 ± 0 | 0 ± 0 | 0 ± 0 | 0 ± 0 | 0 ± 0 | 0 ± 0 | 0 ± 0 |
| Phylum | 0 ± 0 | 7 ± 3 | 0 ± 0 | 0 ± 0 | 13 ± 1 | 0 ± 0 | 0 ± 0 | 13 ± 1 | 0 ± 0 |
| Class | 1 ± 1 | 10 ± 2 | 1 ± 1 | 1 ± 0 | 19 ± 1 | 1 ± 0 | 1 ± 0 | 21 ± 2 | 1 ± 0 |
| Order | 7 ± 2 | 33 ± 1 | 8 ± 2 | 17 ± 0 | 35 ± 1 | 22 ± 1 | 12 ± 1 | 41 ± 1 | 14 ± 2 |
| Family | 35 ± 10 | 69 ± 8 | 34 ± 10 | 34 ± 1 | 62 ± 1 | 37 ± 1 | 28 ± 2 | 60 ± 2 | 28 ± 2 |
| Genus | 55 ± 9 | 81 ± 7 | 55 ± 8 | 63 ± 1 | 73 ± 1 | 63 ± 1 | 56 ± 1 | 71 ± 2 | 55 ± 2 |
| Species | 98 ± 1 | 100 ± 0 | 100 ± 0 | 99 ± 0 | 100 ± 0 | 100 ± 0 | 99 ± 0 | 100 ± 0 | 99 ± 0 |

Data (means ± standard deviation (n = 4)) were determined with the three data analyses tools CoMA, Mothur and QIIME. All values are presented in percent of total reads of each sample.

was included to compare the OTU-based results also with an ASV approach. ASVs are currently proposed as an alternative to OTUs [28, 78] and we believe that a proper validation of CoMA would not be possible if neglecting this progression. For the real dataset, however, analyses focused solely on OTU-based strategies, which are still predominant in soil microbiological studies.

**Benchmarking with mock communities.** The benchmark test showed that all tested pipelines (CoMA, Mothur, QIIME, QIIME2) were able to depict the taxonomy in each of the three mock communities in a proper way. Irrespective of the dataset, taxa were found with a high accuracy and a relatively low error rate. However, when looking closer, differences in performance were found among the tools depending on the dataset. While the CoMA output was closely related to that of QIIME2 in *mock-13* and *mock-26*, a considerable distance between these two tools was seen for *mock-16*, where CoMA was almost identical with QIIME. This indicates that the accuracy of an analysis highly depends on the dataset and that no general conclusion for the best performing analysis tool can be drawn when not considering sample specificities. Decisive characteristics of input data may include the applied primer system (e.g. 16S rRNA, 18S rRNA, ITS, functional genes), the variable region (e.g. V3 or V4 for 16S rRNA), the sequence length, as well as the taxonomic structure of the sample (e.g. species richness, evenness). Even though a general conclusion is difficult, CoMA seemed to be the overall most consistent platform in this test set since it did not show any severe inaccuracies for any of the three datasets, while the other tools did so for at least one mock community.

When comparing the OTU-based platforms (CoMA, Mothur, QIIME) with QIIME2 computing ASVs, no clear trend was observed. For *mock-26*, the fungal ITS dataset, QIIME2 performed particularly well and was the most accurate tool for this dataset in terms of mean per-taxon deviation. Overall, however, QIIME2 was ranked last, what may be explained with the averagely lowest genus detection rate ($< 83\%$) among all platforms within the test set, especially seen for *mock-16*. Here, QIIME2 was not able to reveal almost 25% (12 out of 46) of all bacterial/archaeal genera. These results evoke the question whether ASVs are truly preferable over OTUs, albeit it needs to be emphasized that this study was not designed to compare OTUs and ASVs in all its aspects and hence no general conclusion can be drawn. Callahan et al. [78] suggested a general replacement of OTUs in favor of ASVs for all marker-gene analyses and argued with a higher reproducibility, a finer sequence resolution and the advantage of comparability between different studies. On the other hand, existing sequencing technologies often lack precision and hence an accurate resolving of exact sequences appears doubtful. Choosing the clustering strategy is a tradeoff between lumping taxa on the one hand (OTUs) and splitting them on the other hand (ASVs), and the decision seems to depend on the analyzed dataset, particularly in terms of the applied primer system and the length of the inner primer region. This is also supported by Edgar [79], who tested various OTU cutoff levels and found optima of 99% for full length 16S rRNA, and 100% for short hypervariable regions (e.g. V4 of the 16S rRNA). For the next years, we expect a co-existence of OTUs (probably with a 99% cutoff instead of 97%) and ASVs, always depending on the data to analyze. However, as soon as the sequencing technologies are reaching the next level in terms of precision, ASVs may be favorable and the field should clearly move in this direction.

Irrespective of the dataset, the results of all pipelines were more closely related to each other than to the expected values (with one exception: QIIME at *mock-26*). This indicates that the inaccuracies in this benchmark test were, to a certain degree, caused by the sequencing process rather than by the applied analysis tool. Several other authors who reported a systematic distortion of NGS data [80, 81] drew the same conclusion. Biases occur all over the process, starting with the extraction of DNA, the PCR, and the sequencing procedure itself [82]. According to the authors, the steps of DNA extraction and PCR are particularly susceptible to errors,

which further accumulate until the end of the sequencing process. This results in a severe bias towards some strains, while others are neglected [83]. These aspects always need to be considered when using amplicon sequencing and we strongly suggest to include samples with a known composition (= mock community) to the analysis in order to estimate the degree of introduced inaccuracy.

In all test categories, CoMA satisfied with a good performance in the benchmark test based on mock communities. It seems to provide proper results, irrespective of the characteristics of the analyzed dataset. The CoMA pipeline is therefore suggested as highly accurate alternative to the established NGS analysis platforms for any kind of input data. Future CoMA releases will also include the possibility to select the OTU cutoff level (currently: 97%) and to generate ASVs rather than OTUs (using the UNOISE3 algorithm [84]), making the pipeline more flexible and congruent with different opinions in the field no matter in which direction sequence analysis is heading in the future.

**Platform comparison using real soil data.** Comparing the three soils, all of the three pipelines used for the soil dataset (CoMA, Mothur, QIIME) generally showed the same trends concerning microbial diversity. The overall level of diversity, however, was different among these analysis tools, which can be explained by the way diversity and richness are typically calculated. Most algorithms, also those in CoMA, are based on OTUs. Therefore, OTU clustering and subsequent data post-processing steps are crucial and may significantly affect diversity calculations, as well as all other upstream analyses. Generally, there are three different strategies for clustering sequences into OTUs: closed-reference OTU picking, *de novo* assembling and open-reference OTU picking [72]. Closed-reference OTU picking first assigns sequences to a taxonomic database and thereafter clusters them based on the taxonomic classification. This process is fast, but some problems arise when taxonomic assignment is ambiguous or when no similar entry can be found in the reference database [85]. *De novo* assembling reverses this approach and clusters sequences in a first step, followed by the taxonomic assignment. This strategy is more robust, especially when analyzing complex datasets, but also tends to be more intense in terms of computational effort [86]. Open-reference clustering manifests itself a hybrid of the above approaches. Similar to closed-reference OTU picking, sequences are first assigned to a reference database and unclassified reads are thereafter assembled *de novo*. This third option combines advantages of both approaches; however, it leads to two different OTU definitions within one single strategy [87].

CoMA as well as Mothur and QIIME all used *de novo* assembling for clustering of the sequencing data on a 97% similarity level. However, each tool implements individual algorithms or rather complete tool packages. CoMA and QIIME use USEARCH (in fact UPARSE [54]) for OTU clustering by using different versions of this algorithm (CoMA: USEARCH 10.0.240, QIIME: USEARCH 6.1.544), though. Mothur uses OptiClust [86] instead. Assuming that both USEARCH versions used in CoMA and QIIME are largely comparable, particularly OptiClust may have caused such differences with regard to Mothur. This is supported by our data where CoMA and QIIME were generally more closely related to each other than to Mothur. Beyond the fundamental OTU clustering, each pipeline includes various steps for quality filtering, including data demultiplexing, data denoising and chimera removal. All these steps vary among the tools and may significantly affect the results of data analyses. Subsequent removal of rare OTUs as well as data subsampling would evoke different outputs as well. However, this can be disregarded for the present comparison since all OTU-tables were treated with the same algorithms.

Data analysis with Mothur resulted in different microbial compositions in the three soil habitats even though most of the key microbes were found with all of the three pipelines. Despite the different algorithms for OTU clustering and data quality filtering, the output of

these analyses might be strongly influenced by the aligning tool/mechanism and the used reference database for taxonomic assignment. In all cases, alignment was done via different tools: BLAST in CoMA, RDP in Mothur and UCLUST in QIIME, following the recommendations in the respective SOPs. However, it should be borne in mind that all of the three pipelines also offer alternative aligners and as such, we are not addressing this aspect in detail. Taking a closer look at the results revealed that most of the conflicting taxa showed particularly low abundances. For the composition of low-abundant taxa, removal of rare reads has a particularly high impact. The same taxon may be slightly above the cutoff in one analysis, whereas it is slightly below in the other. Consequently, a taxon that was in fact determined with both tools may appear just in one of them, resulting in a decreased coverage. We assume that the applied reference database is another important driving factor for the varying results between CoMA/ QIIME and Mothur. Both, CoMA and QIIME used the newest distribution of SILVA (SLV_132). Taxonomic classification with Mothur, however, was done using the current release of the RDP trainset (Version 16) as described in the SOP. This explains the differences found in terms of unclassified taxa since SILVA includes significantly more sequences than RDP [88], resulting in a better coverage. Nevertheless, this does not explain the mismatches when comparing microbial core families. These taxa were not only different at family level but also down to phylum or class.

Taken together, all comparisons indicate that the CoMA output is generally well comparable to that of Mothur and QIIME, particularly when looking at the microbial key players. Regarding less abundant taxa, differences to Mothur were found.

## Assets of the CoMA pipeline

The main advantage of CoMA is the intuitive and user-friendly operation that is supported by the graphical user interface. It allows entry-level users to perform amplicon-sequencing data analysis and to obtain solid results without the need for time- and effort-consuming training. Beginners are further supported by the detailed CoMA manual and a systematic tutorial on basis of a provided simple model dataset. All required steps and input parameters are explained in detail and the small size of the dataset ($< 1250$ reads per sample) allows for a fast progress without long downtimes. Advanced users can utilize the provided high degree of automation to analyze huge amounts of data efficiently. Moreover, they find various settings for optimizing the workflow and adapting the analysis specifically to their needs. CoMA also offers the possibility to stop an uncompleted analysis at any point to continue later or to recalculate parts of the workflow. This allows for the improvement of a former run by simply adjusting decisive input parameters without the need for a complete recalculation. As an open-source tool package, CoMA's source code may even be adapted or expanded according to the needs of ambitious users.

CoMA offers a huge variety of functions, starting with data pre-processing and ending with several useful tools for data visualization and statistical appraisal. Data pre-processing is following a state of the art procedure for amplicon-sequencing data: merging of paired-end reads, trimming of primers/ barcodes/adapters and quality control with the possibility to filter bad reads. Also OTU clustering (and in the future ASV clustering), taxonomic assignment and data post-processing are done according to the current state of knowledge. A particular focus of CoMA was laid on graphics that meet both esthetic and scientific standards, appropriate for publication. The graphics can be created in many different file formats and pixel depths in order to fulfill specific journal requirements. Taxa plots can be created as bar charts but also as heatmaps. The latter are convenient for the depiction of taxonomic data and we strongly encourage them particularly for big datasets. Venn plots are another option for the illustrative

comparison of taxonomic data. Two or three specifically selected groups can be compared with each other in terms of shared taxa, available for each taxonomic level. In future CoMA releases, the possibility for comparing four or five groups shall be implemented. However, Venn diagrams including too many groups tend to be confusing and lose the clarity of its original idea. Alpha- and beta diversity (ordination, cluster analysis) can be calculated using multiple methods and metrics, always providing the optimal combination for the respective dataset.

CoMA also provides the results as text files in standardized formats, which can be used for additional or more sophisticated analyses by advanced users. This may include for instance statistical analyses with software packages such as R. Moreover, taxonomic trees can be created using a text file in NEWICK format. This may either end in a basic depiction of the tree structure (e.g. FastTree [89]) or in highly sophisticated and complex circular plots (e.g. GraPhlAn), combining the taxonomic tree with circular bar charts or heatmaps. The CoMA output can also be used for the determination of overrepresented taxa (e.g. LEfSe, IndVal [90]), the creation of taxonomic networks (e.g. Cytoscape) or the construction of pseudo-metagenomic data to reveal the connection between taxonomy and functional genes (e.g. PICRUSt, Tax4Fun [91], Piphillin [92]). These are only some examples for further usage, and the choice depends on the research question and the dataset.

CoMA can be run on every common computer operating system (e.g. Linux, Windows, macOS). Currently, three different options for installation are available: a virtual appliance (which can be imported with tools like VMware Workstation, Oracle VM Virtualbox or Parallels Desktop for Mac), a Singularity image, and a direct Linux installer. Each option provides specific advantages and the user should select the most suitable one in order to meet his needs. The CoMA manual includes detailed information for each installation option and describes it thoroughly. When choosing the Singularity option, CoMA can be easily used on high performance computer (HPC) systems. Working on HPC clusters is becoming more and more important in modern NGS data analysis, and applying CoMA here combines easy and intuitive operation with tremendous computing power.

Concluding, CoMA offers several advantages over currently used pipelines, especially for beginners but also for advanced users. This includes particularly the intuitive and user-friendly operation, the flexible usage on any operating system as well as a graphical output in publication-ready form. We strongly recommend the usage of CoMA for convenient and efficient NGS data analysis for researchers, both in the biological and medical field. CoMA will be updated regularly to ensure an excellent performance also in the future, but also to implement additional utilities (e.g. ASV support). These updates will always include the newest versions of the taxonomic databases at the release date to guarantee the best possible results.

## Supporting information

**S1 Fig. Rarefaction-curve analysis based on operational taxonomic units (OTUs) created with the CoMA pipeline to evaluate the sequencing effort.** F = forest. GR = grassland. S = swamp.
(TIF)

**S2 Fig. Hierarchical cluster analysis showing the cosine similarity between four different analysis platforms and the set point (SP) for the *mock-13* dataset.** The dendrogram was calculated with the UPGMA method (unweighted pair group method with arithmetic mean) as bottom-up approach.
(TIF)

**S3 Fig. Hierarchical cluster analysis showing the cosine similarity between four different analysis platforms and the set point (SP) for the *mock-16* dataset.** The dendrogram was calculated with the UPGMA method (unweighted pair group method with arithmetic mean) as bottom-up approach.
(TIF)

**S4 Fig. Hierarchical cluster analysis showing the cosine similarity between four different analysis platforms and the set point (SP) for the *mock-26* dataset.** The dendrogram was calculated with the UPGMA method (unweighted pair group method with arithmetic mean) as bottom-up approach.
(TIF)

**S5 Fig. Hierarchical cluster analysis showing the cosine similarity between four different analysis platforms and the set point (SP) for all mock dataset (*mock-13*, *mock-16*, *mock-26*).** The dendrogram was calculated with the UPGMA method (unweighted pair group method with arithmetic mean) as bottom-up approach.
(TIF)

**S6 Fig.** (A) Abundance (OTU, operational taxonomic unit), (B) Simpson diversity (D) and (C) Chao1 diversity of three different soils after sequencing data analysis with CoMA, Mothur and QIIME. Four replicates are shown for each habitat. F = forest. GR = grassland. S = swamp.
(TIF)

**S1 Table. Physico-chemical soil properties of three sites in Gschnitz valley, Tyrol (Austria) investigated to validate the CoMA pipeline.** The table shows means ± standard deviation (n = 4). DM = dry matter. VS = volatile solids. EC = electrical conductivity. WHC = water holding capacity.
(DOCX)

**S1 File. Soil data analysis with CoMA.** The file includes the results and a discussion of the microbial characteristics of three different soils (forest, grassland, swamp), which was determined with 16S amplicon sequencing.
(PDF)

## Acknowledgments

We thank Philipp Dresch for his much-appreciated help in terms of sample collection and preparation, and Monika Leitner for giving us the permission to take soil samples from her property. Moreover, we thank Florian Steiner for critical comments on an earlier version of the manuscript, and Maraike Probst as well as Thomas Klammsteiner for helpful advice in terms of data analysis with Mothur.

## Author Contributions

**Conceptualization:** Sebastian Hupfauf, Mohammad Etemadi, Heribert Insam, Sabine Marie Podmirseg.

**Data curation:** Sebastian Hupfauf.

**Formal analysis:** Sebastian Hupfauf.

**Funding acquisition:** Heribert Insam, Sabine Marie Podmirseg.

**Investigation:** Sebastian Hupfauf, Marina Fernández-Delgado Juárez, María Gómez-Brandón.

**Methodology:** Sebastian Hupfauf, Mohammad Etemadi.

**Project administration:** Sebastian Hupfauf, Heribert Insam, Sabine Marie Podmirseg.

**Resources:** Sebastian Hupfauf, Heribert Insam, Sabine Marie Podmirseg.

**Software:** Sebastian Hupfauf, Mohammad Etemadi.

**Supervision:** Heribert Insam, Sabine Marie Podmirseg.

**Validation:** Sebastian Hupfauf.

**Visualization:** Sebastian Hupfauf.

**Writing – original draft:** Sebastian Hupfauf.

**Writing – review & editing:** Sebastian Hupfauf, Mohammad Etemadi, Marina Fernández-Delgado Juárez, María Gómez-Brandón, Heribert Insam, Sabine Marie Podmirseg.

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
