## [Decision Letter · Decision Letter 0]

2 Sep 2020

PONE-D-20-21940

CoMA – an Intuitive and User-friendly Pipeline for Amplicon-Sequencing Data Analysis

PLOS ONE

Dear Mr. Hupfauf,

Thank you for submitting your manuscript to PLOS ONE. After careful consideration, we feel that it has merit but does not fully meet PLOS ONE’s publication criteria as it currently stands. Therefore, we invite you to submit a revised version of the manuscript that addresses the points raised during the review process.

We look forward to receiving your revised manuscript.

Kind regards,

Zechen Chong

Academic Editor

PLOS ONE

Journal Requirements:

Reviewers' comments:

Reviewer's Responses to Questions

**Comments to the Author**

1. Is the manuscript technically sound, and do the data support the conclusions?

Reviewer #1: Yes

Reviewer #2: Partly

Reviewer #3: Yes

2. Has the statistical analysis been performed appropriately and rigorously? 

Reviewer #1: Yes

Reviewer #2: Yes

Reviewer #3: Yes

3. Have the authors made all data underlying the findings in their manuscript fully available?

Reviewer #1: Yes

Reviewer #2: Yes

Reviewer #3: Yes

4. Is the manuscript presented in an intelligible fashion and written in standard English?

Reviewer #1: Yes

Reviewer #2: Yes

Reviewer #3: Yes

5. Review Comments to the Author

Reviewer #1: The manuscript by Hupfauf et al. describes COMA, a pipeline designed to the analysis of metabarcoding data. The paper is overall well written. However, there are some flaws that need to be addressed as illustrated below.

Introduction:

L77-78 The list of reference databases should be improved by including MIDORI, PR2 DB and ITSoneDB. Moreover authors should discuss the fact that actually both Greengenes and RDP are outdated.

L 88. Authors should discuss the differences between OTU and ASV. They cite the paper by Callahan et al 2017 in Discussion but they did not describe the technical differences between the two approaches.

L 126 454 was terminated in 2016.

Table 1: Add PEMA

M&M

L 170-172 Authors stated in COMA a denoising and error correction step is implemented by using Lotus. It is not clear what they mean with “denoising and error correction”. Literally denoising is the procedure applied to remove the noise introduced by amplification and sequencing, as stated by Quince et al in the paper describing AmpliconNoise. Actually, the denoising procedures are strictly related to the used sequencing platform as well as different sequencing approaches are characterized by different error profiles. In the introduction, authors declare COMA is able to analyze Illumina, PacBio and Roche 454 data, consequently if a denoising step is implemented in the pipeline, it must take into account the different peculiarities of these sequencing approaches. Authors should clarify this point.

L 176 The RDP-classifier is a Bayesian classifier inferring the posterior probability based on the known kmer distribution in genera and the observed one. Consequently it do not perform any alignment. Moreover it is not described how blast alignment is performed (default options?) and if it is filtered according to the observed similarity.

L 230-343 My major concerns are related to both the choice and the analysis of Mock data. Concerning the selected mocks: i) the inclusion of a mock with few dominant taxa and other less abundant taxa could recapitulate a real condition in several cases; ii) in consideration that 454 has been discontinued the choice of a ITS mock obtained by 454 sequencing is not very useful nowadays. In the following some comments concerning the analysis of mock data. First of all by taking into account for each mock the mixed strains are known, it is possible to infer the expected amplicon sequences, by performing an in silico PCR on the reference genomes. This would also allow to measure the intra- and inter- generic sequence variability and compare the observed OTUs/ASVs with the expected sequences. Moreover, it is not completely clear how taxonomic assignment was performed with the different pipelines. While in COMA an alignment based approach was used (see comments above for related issues), in QIIME the author used the assign-taxonomy.py script which implements 5 different taxonomic assignment methods (including blast), but they do not indicate the used one (uclust is the default). The same issue is related to the taxonomic classification in Mothur while in QIIME2 a naïve-bayesian classifier trained on the same target region was used. Actually it seems different methods for taxonomic classification were applied and this may influence the comparison of the results. Moreover, the ASV inference pipeline implemented in QIIME2 is deeply influence by the used filtering parameters. Have the authors used the default parameters or have adjusted them according to the overall data quality? Finally, authors should indicate the versions of the used tools.

Figure 2, 3 and 4 I suggest to change the colors in order to improve the plots.

In general, I find the manuscript is too long and not focused. Indeed, it includes two messages: the description of a novel pipeline for DNA metabarcoding analysis, and an analysis of real data of soil samples. Maybe the Authors can consider to reduce the part concerning real data to the comparative aspects, moving the microbiome analysis of soil samples and its discussion to a different manuscript focused on this matter.

Reviewer #2: This manuscript reported a pipeline CoMA for amplicon-sequencing data analysis. It reported that comparing to other command line tools, the pipeline CoMA's advantages are much easier, more user-friendly, compatible with any common operating system, and good performance compared.

Major comments:

1. The CoMA is distributed by a 5.77GB VirtualBox .ova file. I cannot test it on my remote lab server because it doesn't have GUI to use VirtualBox. Due to COVID19, I couldn't access my high performance desktop computer in my office. I couldn't test it on my home laptop because of CoMA's large disk space requirement (50GB). Hence, I am sorry that I didn't test whether it can be the installed and work correctly. Hope other reviewers did that. In fact, I don't think it's an advantage to use virtual machine image for distributing CoMA. I think Docker is a much better and easier solution for both software maintainers and users.

2. As I didn't run CoMA by myself, so I didn't see it's graphical interface. However, according to the description in the manuscript and CoMA's manual online, I don't think its GTK+ dialog GUI is better than a command line interactive mode. I don't think command line is hard to use, and I don't think CoMA is easier and user-friendly than command line tools.

3. It reported that "the workflow results in highly esthetic and publication-ready graphics". However, I didn't find any example figure in the manuscript and CoMA's website. As I didn't run CoMA, so I cannot check how good the output figures are. I think the authors should provide a demo/tutorial or some example output figures on their website.

4. It reported that "we used various open-source, third-party tools and combined them into a linear analysis workflow in the form of a Bash script". Hence, I think the core of CoMA is just a bash script. I think the authors need to make their code available (maybe on GitHub) if CoMA is an open source software.

5. In addition, the authors didn't describe which tools are used in CoMA and in which steps they are used. In Figure 1, the main tools used in each step need to be marked.

6. It reported that the results generated by CoMA and QIIME were very similar. Is it because CoMA used QIIME in several key steps?

7. For Soil dataset, QIIME2 wasn't included in the comparison. Why?

Minor comments:

1. Line 126, delete the space in the URL.

Reviewer #3: The authors present a novel practical pipeline for processing of next-generation datasets for microbiome analyses. The modules in this pipelines range from QC to taxonomic assignment and clustering analysis. The pipeline also provides visualization and statistical analysis tools. The pipeline is bench-marked on multiple microbiome datasets such as soil, mock datasets. The experiments suggest that it outperforms other known methods in the field.

The paper is well-written and is a good contribution to the community. A few minor concerns are:

1) A figure representing more details of the tasks (QC, taxonomic assignment) will be helpful.

2) One feature that is encouraged in the pipeline is to compare and validate across technologies.

3) With upcoming HiFi technology, there is a huge opportunity to perform strain-aware metagenome assembly. The computational methods like SDip (please cite https://doi.org/10.1101/2020.02.25.964445) can be explored for assembling microbiomes. This method has the potential to use variation from multiple strains and species to bin reads and perform assembly. A few sentences on third-generation technologies and analysis methods in the manuscript are appreciated.

6. PLOS authors have the option to publish the peer review history of their article (what does this mean?). If published, this will include your full peer review and any attached files.

Reviewer #1: No

Reviewer #2: No

Reviewer #3: No

---

## [Author Response · Author response to Decision Letter 0]

18 Sep 2020

PONE-D-20-21940

Reviewers' comments

Below are the responses to the Reviewers’ comments. The line numbers we are referring to are those of the revised manuscript (“Revised Manuscript with Track Changes.docx”).

Academic Editor

DONE. We conscientiously revised the manuscript in order to meet the journal’s style requirements. Below, there is a summary of the most important changes:

• Title changed to sentence chase

• Removed telephone and fax numbers of the corresponding author

• Removed the chapter “Author summary”

• Removed the labelling of the headings

• The term “supplementary material” was removed when referencing supplementary figures and tables

• Removed the chapter “Supplementary Material”

 

Reviewer #1

The manuscript by Hupfauf et al. describes COMA, a pipeline designed to the analysis of metabarcoding data. The paper is overall well written. However, there are some flaws that need to be addressed as illustrated below.

Thank you for sharing your constructive feedback! The performed changes based on your suggestions have truly increased the quality of this manuscript!

Introduction:

L77-78 The list of reference databases should be improved by including MIDORI, PR2 DB and ITSoneDB. Moreover authors should discuss the fact that actually both Greengenes and RDP are outdated.

DONE. We have included the three additional databases, and provided now the year of the last update for the Greengenes and RDP databases in order to clarify that these databases are not up to date anymore; please see lines 68-71.

L 88. Authors should discuss the differences between OTU and ASV. They cite the paper by Callahan et al 2017 in Discussion but they did not describe the technical differences between the two approaches.

DONE. We have compared OTU and ASV approaches now in more detail, please see lines 76-81.

L 126 454 was terminated in 2016.

DONE. Changed accordingly, please see line 116.

Table 1: Add PEMA

DONE. We have added the PEMA pipeline to Table 1.

M&M

L 170-172 Authors stated in COMA a denoising and error correction step is implemented by using Lotus. It is not clear what they mean with “denoising and error correction”. Literally denoising is the procedure applied to remove the noise introduced by amplification and sequencing, as stated by Quince et al in the paper describing AmpliconNoise. Actually, the denoising procedures are strictly related to the used sequencing platform as well as different sequencing approaches are characterized by different error profiles. In the introduction, authors declare COMA is able to analyze Illumina, PacBio and Roche 454 data, consequently if a denoising step is implemented in the pipeline, it must take into account the different peculiarities of these sequencing approaches. Authors should clarify this point.

The quality filtering is done using LotuS, precisely the sdm tool. Regarding the LotuS publication (https://doi.org/10.1186/2049-2618-2-30), sdm “… filter(s) input sequences after average quality, accumulated error over the sequence, quality in a freely definable window and remove 5′ low-quality bases filtered for these criteria. Furthermore, sequences are filtered for min/max length, ambiguous nucleotides, max barcode and primer errors, polynucleotide runs, and trimmed for adapter sequences, if present.”. The reviewer is right in the sense that the filtering settings largely depend on the sequencing platform; hence, LotuS provides pre-adjusted sdm setting files for MiSeq, HiSeq, 454 and PacBio. The newest version of the CoMA pipeline now asks the user, which sequencing platform was used, and selects the corresponding sdm file for the sequence filtering process.

L 176 The RDP-classifier is a Bayesian classifier inferring the posterior probability based on the known kmer distribution in genera and the observed one. Consequently it do not perform any alignment. Moreover it is not described how blast alignment is performed (default options?) and if it is filtered according to the observed similarity.

DONE. We slightly rephrased this sentence in order to prevent any misunderstandings, please see lines 169-172. Blast is performed using the LotuS package with the default LotuS options. More details can be found in the LotuS publication (https://doi.org/10.1186/2049-2618-2-30).

L 230-343 My major concerns are related to both the choice and the analysis of Mock data. Concerning the selected mocks: i) the inclusion of a mock with few dominant taxa and other less abundant taxa could recapitulate a real condition in several cases; ii) in consideration that 454 has been discontinued the choice of a ITS mock obtained by 454 sequencing is not very useful nowadays.

Thank you very much for your feedback on our Mock analysis! The selection of suitable Mock datasets was done according to the recommendations of a previous reviewer. We tried to cover as much variation as possible in terms of diversity, marker genes, and sequencing platforms. By doing that, we wanted to demonstrate how CoMA performs when analyzing different types of data, always in comparison to established tools. It would indeed be interesting to analyze much more different Mock datasets (e.g. highly uneven distributed taxa, functional gene markers, PacBio data, etc.). However, this additional information would further extend the scope of this MS and it would be better to address it within another publication.

In the following some comments concerning the analysis of mock data. First of all by taking into account for each mock the mixed strains are known, it is possible to infer the expected amplicon sequences, by performing an in silico PCR on the reference genomes. This would also allow to measure the intra- and inter- generic sequence variability and compare the observed OTUs/ASVs with the expected sequences.

This is indeed an interesting approach! We are gladly taking this idea for our upcoming study and expect to gain even more information out of our analysis, particularly in terms of sequence variability.

Moreover, it is not completely clear how taxonomic assignment was performed with the different pipelines. While in COMA an alignment based approach was used (see comments above for related issues), in QIIME the author used the assign-taxonomy.py script which implements 5 different taxonomic assignment methods (including blast), but they do not indicate the used one (uclust is the default).

DONE. In QIIME, taxonomic assignment was done with “uclust”. In order to avoid any uncertainty, we have added this information; please see lines 301-303.

The same issue is related to the taxonomic classification in Mothur while in QIIME2 a naïve-bayesian classifier trained on the same target region was used. Actually it seems different methods for taxonomic classification were applied and this may influence the comparison of the results.

We are aware of the different algorithms that are implemented in the compared tools. Our approach, however, was to compare CoMA with the suggested procedures (SOPs) of Mothur, QIIME and QIIME2. In the end, all tools should depict the mock community as precisely as possible, irrespective of the applied algorithms (what they did quite well).

Moreover, the ASV inference pipeline implemented in QIIME2 is deeply influence by the used filtering parameters. Have the authors used the default parameters or have adjusted them according to the overall data quality? Finally, authors should indicate the versions of the used tools.

DONE. As for QIIME and Mothur, we also used the recommended settings for the QIIME2 analysis from the official SOP documentation. We are providing the applied versions of Mothur, QIIME and QIIME2 now in the manuscript; please see lines 260, 292 and 311.

Figure 2, 3 and 4 I suggest to change the colors in order to improve the plots.

Figures 2-4 show bar charts using the color code that is currently used by CoMA for all graphics. Hence, we would like to keep the colors as they are, in order to demonstrate the CoMA style continuously throughout the whole paper. However, we are aware that sometimes users might have very specific color-requirements. Therefore, we have decided to implement an option for selecting an individual color map for all plots in CoMA within the next update of the pipeline.

In general, I find the manuscript is too long and not focused. Indeed, it includes two messages: the description of a novel pipeline for DNA metabarcoding analysis, and an analysis of real data of soil samples. Maybe the Authors can consider to reduce the part concerning real data to the comparative aspects, moving the microbiome analysis of soil samples and its discussion to a different manuscript focused on this matter.

Thank you for this critical comment! We had discussed about this issue when starting to write the paper. After weighing the pros and cons of combining a method paper with the analysis of real data, we decided to keep it combined because we are convinced that it is beneficial for the majority of CoMA users to see the pipeline being demonstrated on a real dataset (including an extensive interpretation and discussion of the results). Moreover, we aimed at providing any relevant information on CoMA in a single extensive publication rather than splitting it up into two separate papers, one solely describing the pipeline and one demonstrating its functionality based on real data. 

Reviewer #2

This manuscript reported a pipeline CoMA for amplicon-sequencing data analysis. It reported that comparing to other command line tools, the pipeline CoMA's advantages are much easier, more user-friendly, compatible with any common operating system, and good performance compared.

Thank you very much for reviewing our manuscript and for sharing your constructive feedback! Please find below our detailed response to each individual point.

Major comments:

1. The CoMA is distributed by a 5.77GB VirtualBox .ova file. I cannot test it on my remote lab server because it doesn't have GUI to use VirtualBox. Due to COVID19, I couldn't access my high performance desktop computer in my office. I couldn't test it on my home laptop because of CoMA's large disk space requirement (50GB). Hence, I am sorry that I didn't test whether it can be the installed and work correctly. Hope other reviewers did that. In fact, I don't think it's an advantage to use virtual machine image for distributing CoMA. I think Docker is a much better and easier solution for both software maintainers and users.

It is unfortunate that you were not able to test CoMA personally! Please feel free to test it later, your feedback is highly appreciated! We are currently working on alternative ways of installing CoMA in the near future. This will include a Docker image, a direct Linux installer, as well as a Singularity container, which will also allow the usage of CoMA on a HPC cluster system. We are very confident to provide all of them within the next two or three weeks and will advertise these novel features on the CoMA homepage.

2. As I didn't run CoMA by myself, so I didn't see it's graphical interface. However, according to the description in the manuscript and CoMA's manual online, I don't think its GTK+ dialog GUI is better than a command line interactive mode. I don't think command line is hard to use, and I don't think CoMA is easier and user-friendly than command line tools.

CoMA was initially coded as command line-based pipeline. After having been extensively tested (including by entry-level users), we decided to include the GTK+ dialogs taking their feedback into account. According to our testers, this considerably facilitated the workflow. Apart from that, there are also other aspects making CoMA user-friendly and easy to use for beginners. We are providing a detailed manual covering the installation of the tools as well as the complete analysis of a test data set. The test data set is already available inside CoMA and can be analyzed without the need to download it first from a repository. Each step of the workflow is explained in detail and additional literature is provided where it might be useful. Our experience is that especially entry-level users often struggle with solely command-line-based tools. Nevertheless, they can consult the source code any time and learn its’ usage step by step.

3. It reported that "the workflow results in highly esthetic and publication-ready graphics". However, I didn't find any example figure in the manuscript and CoMA's website. As I didn't run CoMA, so I cannot check how good the output figures are. I think the authors should provide a demo/tutorial or some example output figures on their website.

DONE. Thank you for this great suggestion! We have included a demo gallery showing the different graphics that can be created with CoMA to our webpage (https://www.uibk.ac.at/microbiology/services/coma.html). Certainly, this will help to get a better idea of CoMA before installing it.

4. It reported that "we used various open-source, third-party tools and combined them into a linear analysis workflow in the form of a Bash script". Hence, I think the core of CoMA is just a bash script. I think the authors need to make their code available (maybe on GitHub) if CoMA is an open source software.

CoMA is implemented as Bash script calling both, already available open-source tools as well as in-house Python scripts. Particularly the steps for data visualization and statistical appraisal are mainly done using our own scripts. We will provide all code on a public repository (GitHub) upon official publication of the CoMA pipeline. The information on the access to this code is now given in the manuscript; please see line 871.

5. In addition, the authors didn't describe which tools are used in CoMA and in which steps they are used. In Figure 1, the main tools used in each step need to be marked.

All third party tools are provided in the “Implementation of CoMA” section (2.1.) of the manuscript. A complete summary is also included in the CoMA manual (chapter “Software list”). However, we did not include this information to Figure 1 since most steps are done by a combination of third party tools and CoMA scripts. Moreover, we wanted to keep the depiction of the workflow as concise as possible. 

6. It reported that the results generated by CoMA and QIIME were very similar. Is it because CoMA used QIIME in several key steps?

This is not the fact, QIIME scripts are only used in two steps within CoMA, namely removal of rare OTUs and subsampling. Both steps are called after the creation of the OTU table and no major effect on the overall results is expected. Moreover, the removal of rare OTUs and subsampling was also done with the same QIIME scripts on the OTU tables originating from Mothur, QIIME and QIIME2.

7. For Soil dataset, QIIME2 wasn't included in the comparison. Why?

For the soil data analysis, we decided to focus on OTU-based approaches since the majority of current soil studies still compares OTUs. Moreover, we wanted to focus clearly on the biological interpretation of the data in this section, and believe that tackling also the issue of OTU versus ASV would be beyond the scope here.

Minor comments:

1. Line 126, delete the space in the URL.

DONE.

 

Reviewer #3

The authors present a novel practical pipeline for processing of next-generation datasets for microbiome analyses. The modules in this pipelines range from QC to taxonomic assignment and clustering analysis. The pipeline also provides visualization and statistical analysis tools. The pipeline is bench-marked on multiple microbiome datasets such as soil, mock datasets. The experiments suggest that it outperforms other known methods in the field.

The paper is well-written and is a good contribution to the community. A few minor concerns are:

Thank you very much for your positive feedback, we highly appreciate it! Please find below our individual responses to the points you raised.

1) A figure representing more details of the tasks (QC, taxonomic assignment) will be helpful.

DONE. We have added additional information on the taxonomic assignment step to the caption of Figure 1. The quality control steps in CoMA call the PRINSEQ tool, which produces a quality report for each sample. We have decided to leave this out of Figure 1 since it might not be relevant for most readers at this stage. However, we have included now the additional information in the chapter “Data pre-processing and quality checking”; please see lines 157-159.

2) One feature that is encouraged in the pipeline is to compare and validate across technologies.

I hope we are interpreting this point raised by the reviewer in the right way. Generally, all tested pipelines (CoMA, Mothur, QIIME and QIIME2) performed well and revealed the general pattern of the mock communities correctly. However, there were minor differences between the individual results, which can be explained by the different approaches/algorithms used by each tool. Which pipeline is performing best highly depends on the analyzed dataset and its structure as well as on the applied user settings. We recommend adjusting the selected pipeline individually based on the characteristics of the dataset and always test different settings. Applying a second tool may help validating the results, particularly when analyzing unfamiliar data.

3) With upcoming HiFi technology, there is a huge opportunity to perform strain-aware metagenome assembly. The computational methods like SDip (please cite https://doi.org/10.1101/2020.02.25.964445) can be explored for assembling microbiomes. This method has the potential to use variation from multiple strains and species to bin reads and perform assembly. A few sentences on third-generation technologies and analysis methods in the manuscript are appreciated.

DONE. We are providing some information on third-generation sequencing in the introduction chapter and provide the SDip citation as suggested; please see lines 117-120. However, the focus of this paper is short reads yielded from amplicon sequencing. The processing of long reads and the application of different genome assembling strategies was not covered by our study.

---

## [Decision Letter · Decision Letter 1]

6 Oct 2020

PONE-D-20-21940R1

CoMA – an intuitive and user-friendly pipeline for amplicon-sequencing data analysis

PLOS ONE

Dear Dr. Hupfauf,

Thank you for submitting your manuscript to PLOS ONE. After careful consideration, we feel that it has merit but does not fully meet PLOS ONE’s publication criteria as it currently stands. Therefore, we invite you to submit a revised version of the manuscript that addresses the points raised during the review process.

Please address Reviewer 1 and 2's comments, especially Reviewer 2's Error messages when testing the tool.

We look forward to receiving your revised manuscript.

Kind regards,

Zechen Chong

Academic Editor

PLOS ONE

Additional Editor Comments (if provided):

Please try the best to address Reviewer 1 and 2's comments, especially for reviewer 2's error messages when testing the tool.

Reviewers' comments:

Reviewer's Responses to Questions

**Comments to the Author**

1. If the authors have adequately addressed your comments raised in a previous round of review and you feel that this manuscript is now acceptable for publication, you may indicate that here to bypass the “Comments to the Author” section, enter your conflict of interest statement in the “Confidential to Editor” section, and submit your "Accept" recommendation.

Reviewer #1: (No Response)

Reviewer #2: (No Response)

Reviewer #3: All comments have been addressed

2. Is the manuscript technically sound, and do the data support the conclusions?

Reviewer #1: Yes

Reviewer #2: Partly

Reviewer #3: Yes

3. Has the statistical analysis been performed appropriately and rigorously? 

Reviewer #1: N/A

Reviewer #2: Yes

Reviewer #3: Yes

4. Have the authors made all data underlying the findings in their manuscript fully available?

Reviewer #1: Yes

Reviewer #2: Yes

Reviewer #3: Yes

5. Is the manuscript presented in an intelligible fashion and written in standard English?

Reviewer #1: Yes

Reviewer #2: Yes

Reviewer #3: Yes

6. Review Comments to the Author

Reviewer #1: L 70 ITSoneDB is the correct name

L 71-74 Please clarify what did you mean with “de novo assembling”

L 170-172 Authors stated in COMA a denoising and error correction step is implemented by using Lotus. It is not clear what they mean with “denoising and error correction”. Literally denoising is the procedure applied to remove the noise introduced by amplification and sequencing, as stated by Quince et al in the paper describing AmpliconNoise. Actually, the denoising procedures are strictly related to the used sequencing platform as well as different sequencing approaches are characterized by different error profiles. In the introduction, authors declare COMA is able to analyze Illumina, PacBio and Roche 454 data, consequently if a denoising step is implemented in the pipeline, it must take into account the different peculiarities of these sequencing approaches. Authors should clarify this point.

The quality filtering is done using LotuS, precisely the sdm tool. Regarding the LotuS publication (https://doi.org /10.1186/2049-2618-2-30), sdm “... filter(s) input sequences after average quality, accumulated error over the sequence, quality in a freely definable window and remove 5′ low-quality bases filtered for these criteria. Furthermore, sequences are filtered for min/max length, ambiguous nucleotides, max barcode and primer errors, polynucleotide runs, and trimmed for adapter sequences, if present.”. The reviewer is right in the sense that the filtering settings largely depend on the sequencing platform; hence, LotuS provides pre-adjusted sdm setting files for MiSeq, HiSeq, 454 and PacBio. The newest version of the CoMA pipeline now asks the user, which sequencing platform was used, and selects the corresponding sdm file for the sequence filtering process.

Filtering and denoising are two different things. Based on the authors answer they performs filtering and not denoising. Please fix this point in the manuscript.

Moreover, the ASV inference pipeline implemented in QIIME2 is deeply influence by the used filtering parameters. Have the authors used the default parameters or have adjusted them according to the overall data quality? Finally, authors should indicate the versions of the used tools.

DONE. As for QIIME and Mothur, we also used the recommended settings for the QIIME2 analysis from the official SOP documentation. We are providing the applied versions of Mothur, QIIME and QIIME2 now in the manuscript; please see lines 260, 292 and 311.

The QIIME2 SOP documentation the authors mentions are tutorials designed according to data characteristics and consequently may not work correctly on your data.

Reviewer #2: The authors' responses and revisions addressed some of my comments in the last round of review. However, the most important comments were not well addressed. Generally, I think the core purpose of the manuscript is to introduce the new pipeline CoMA. So I think the comments focusing on the usability of the pipeline are the most important.

1. I suggested using Docker instead of VirtualBox .ova file to distribute the pipeline. I am glad that the authors agreed with my suggestion. They said they will develop "Docker image, a direct Linux installer as well as Singularity container" of CoMA and provide all code on GitHub soon (two or three weeks). However, I think all these things need to be well-prepared and described in the revised manuscript before this submission of revision.

2. It seems that the other two reviewers didn't test the CoMA VM in the last round of review either. Although the CoMA VM would cost a lot of my disk space, I think I should install and test it. However, the usability of CoMA is worse than I expected.

2.1. In the beginning, it couldn't work. I checked the log file and found there is a kind of conflict between my MacOS 10.15.6 and VirtualBox (https://forums.virtualbox.org/viewtopic.php?f=8&t=99739). Then I disabled the mic device in VirtualBox and it worked. Fortunately, it only took 19GB on my Mac instead of the 50GB suggested space.

2.2. Then I followed the CoMA manual to install usearch and then test it with the example data in CoMA VM https://www.uibk.ac.at/microbiology/services/manual-coma.pdf. I still don't agree that the GTK dialog is a user-friendly way. In fact, the CoMA GTK dialog doesn't provide more information than directly display and input in the command line terminal. It's even worse because I couldn't trace back all the parameters I inputted in the GTK dialog window. The pipeline output all the information to the terminal. The authors should redirect log information to separated log files of each tool, and keep the terminal only display a few main information. Besides, as I need to test several times with different choices, so I have to go through the whole dialog windows again and again. If it's a command-line tool, I can just write the whole command once and then copy&paste and modify some parameters. Therefore, I don't agree that GTK dialog is a good way even for entry-level users.

2.3. I ran the pipeline strictly following the manual. It showed that I successfully ran the project. Although I saw several "Error" information in the terminal and the final log file, I thought they were just "warnings", such as "Error: (1431.1) FASTA-Reader: Warning: FASTA-Reader: ".

2.4. However, I tried some other choices and caused some errors such as Python package biom terminated by "UnboundLocalError: local variable data_start referenced before assignment". The CoMA pipeline didn't show me any useful information about whether it's a software problem or caused by a wrong parameter I chose. The funny thing is that the pipeline still showed "Process succeeded!" even though the error happened. In fact, I found the pipeline totally doesn't do exception handling. It's unacceptable.

3. I still think it's necessary to revise Figure 1 to show the main tools or parameters in each main step of the pipeline. In fact, I checked the source code of the pipeline in CoMA VM. Besides the python code for plotting and generating reports, there is not much code for the main steps of the pipeline. Therefore, it shouldn't be a problem to illustrate more detail of the main steps of the pipeline in one figure. And it's worth doing.

4. In fact, I agree with Reviewer #1's comment that the manuscript is too long and not focused. The authors should describe more details of the pipeline, such as the key parameters used in each step, instead of describing the real dataset.

Reviewer #3: (No Response)

7. PLOS authors have the option to publish the peer review history of their article (what does this mean?). If published, this will include your full peer review and any attached files.

Reviewer #1: No

Reviewer #2: No

Reviewer #3: No

---

## [Author Response · Author response to Decision Letter 1]

29 Oct 2020

PONE-D-20-21940R1

Reviewers' comments

Below are the responses to the Reviewers’ comments. The line numbers we are referring to are those of the revised manuscript (“Revised Manuscript with Track Changes.docx”).

Reviewer #1:

L 70 ITSoneDB is the correct name

DONE. Changed accordingly.

L 71-74 Please clarify what did you mean with “de novo assembling”

DONE. We have revised the sentence in order to sort out any uncertainties. Please see lines 67 – 71.

L 170-172 Authors stated in COMA a denoising and error correction step is implemented by using Lotus. It is not clear what they mean with “denoising and error correction”. Literally denoising is the procedure applied to remove the noise introduced by amplification and sequencing, as stated by Quince et al in the paper describing AmpliconNoise. Actually, the denoising procedures are strictly related to the used sequencing platform as well as different sequencing approaches are characterized by different error profiles. In the introduction, authors declare COMA is able to analyze Illumina, PacBio and Roche 454 data, consequently if a denoising step is implemented in the pipeline, it must take into account the different peculiarities of these sequencing approaches. Authors should clarify this point.

“The quality filtering is done using LotuS, precisely the sdm tool. Regarding the LotuS publication (https://doi.org /10.1186/2049-2618-2-30), sdm “... filter(s) input sequences after average quality, accumulated error over the sequence, quality in a freely definable window and remove 5′ low-quality bases filtered for these criteria. Furthermore, sequences are filtered for min/max length, ambiguous nucleotides, max barcode and primer errors, polynucleotide runs, and trimmed for adapter sequences, if present.”. The reviewer is right in the sense that the filtering settings largely depend on the sequencing platform; hence, LotuS provides pre-adjusted sdm setting files for MiSeq, HiSeq, 454 and PacBio. The newest version of the CoMA pipeline now asks the user, which sequencing platform was used, and selects the corresponding sdm file for the sequence filtering process.”

Filtering and denoising are two different things. Based on the authors answer they performs filtering and not denoising. Please fix this point in the manuscript.

DONE. We removed the term “de-noising” in order to avoid any further confusion. Please see lines 166 – 168.

Moreover, the ASV inference pipeline implemented in QIIME2 is deeply influence by the used filtering parameters. Have the authors used the default parameters or have adjusted them according to the overall data quality? Finally, authors should indicate the versions of the used tools.

“DONE. As for QIIME and Mothur, we also used the recommended settings for the QIIME2 analysis from the official SOP documentation. We are providing the applied versions of Mothur, QIIME and QIIME2 now in the manuscript; please see lines 260, 292 and 311.”

The QIIME2 SOP documentation the authors mentions are tutorials designed according to data characteristics and consequently may not work correctly on your data.

DONE. We are sorry for being not sufficiently precise in our response the last time. We followed the QIIME2 SOP in terms of which steps were applied. However, we always considered the sequence quality for determining suitable filtering parameters (e.g. analyzing the output of “qiime demux summarize”). We slightly revised the Material and Methods section describing the QIIME2 data analysis workflow in order to clarify this point; please see lines 318-322.

Addition remark:

Since Reviewer 2 also criticized the length of the manuscript, we have decided to move the parts describing the microbial aspects of the soil samples to the supplementary material (S1 File) as you suggested during the previous round of revision. We believe that this makes the manuscript now better focused on the main purpose of the publication: the presentation and evaluation of the CoMA pipeline. Thank you for this useful suggestion! 

Reviewer #2:

The authors' responses and revisions addressed some of my comments in the last round of review. However, the most important comments were not well addressed. Generally, I think the core purpose of the manuscript is to introduce the new pipeline CoMA. So I think the comments focusing on the usability of the pipeline are the most important.

1. I suggested using Docker instead of VirtualBox .ova file to distribute the pipeline. I am glad that the authors agreed with my suggestion. They said they will develop "Docker image, a direct Linux installer as well as Singularity container" of CoMA and provide all code on GitHub soon (two or three weeks). However, I think all these things need to be well-prepared and described in the revised manuscript before this submission of revision.

DONE. The alternative installation options (Singularity image, Linux installer) are available now on the CoMA webpage (https://www.uibk.ac.at/microbiology/services/coma.html). We updated the manuscript (please see lines 119-121 and 741-748) as well as the manual (chapters ii and iii) in order to describe all installation options. Moreover, all CoMA source files are now uploaded and publicly available on GitHub (https://github.com/SebH87/coma).

2. It seems that the other two reviewers didn't test the CoMA VM in the last round of review either. Although the CoMA VM would cost a lot of my disk space, I think I should install and test it. However, the usability of CoMA is worse than I expected.

2.1. In the beginning, it couldn't work. I checked the log file and found there is a kind of conflict between my MacOS 10.15.6 and VirtualBox (https://forums.virtualbox.org/viewtopic.php

?f=8&t=99739). Then I disabled the mic device in VirtualBox and it worked. Fortunately, it only took 19GB on my Mac instead of the 50GB suggested space.

We are sorry to hear that you encountered problems when installing CoMA. It seems that there is a problem with audio permissions when using VirtualBox on MacOS. This, however, is a VirtualBox issue and we cannot solve it from our side. Nevertheless, we are describing this problem now in the updated CoMA manual in order to making users aware of the problem; please see chapter i.ii. (Notes). Thank you very much for sharing this important issue!

We are glad to provide now also alternative options for installation (please see point 1) so that users can work with CoMA even in such (rare) cases in which VirtualBox is not working properly.

The 50 GB of disk space are just required during the installation (it is the dimension of the virtual hard drive of CoMA). Thereafter, only disk space you are actually using (+ ca. 20 GB for the Linux system and all required software) will be occupied. If you have severe problems with disc space, you should consider using one of the other installation options (e.g. Singularity).

2.2. Then I followed the CoMA manual to install usearch and then test it with the example data in CoMA VM https://www.uibk.ac.at/microbiology/services/manual-coma.pdf. I still don't agree that the GTK dialog is a user-friendly way. In fact, the CoMA GTK dialog doesn't provide more information than directly display and input in the command line terminal.

For sure, both options have their advantages and drawbacks. As already mentioned, we decided to use the GTK dialogs over direct terminal input based on the feedback of several test candidates, most of them entry-level users.

It's even worse because I couldn't trace back all the parameters I inputted in the GTK dialog window. The pipeline output all the information to the terminal.

You can trace all input parameters by checking the log file. Moreover, all parameters are shown in the Terminal window. It could be that you used the old version of CoMA (1.0). This has been improved in the newest version (CoMA 2.0); details where to find this info is given in the newest manual (chapter iv - Tutorial).

The authors should redirect log information to separated log files of each tool, and keep the terminal only display a few main information.

Maybe you used the old version of CoMA (1.0)? Actually, CoMA 2.0 only displays the most important outputs (including parameters set by the user) in the Terminal window. This output is also stored in a log file for each run (xxx.log) so that the users can check all settings later on. All other output (warnings, status messages of third party tools, etc.) is not shown in the Terminal and redirected to a second log file (xxx_detailed.log).

Besides, as I need to test several times with different choices, so I have to go through the whole dialog windows again and again. If it's a command-line tool, I can just write the whole command once and then copy&paste and modify some parameters. Therefore, I don't agree that GTK dialog is a good way even for entry-level users.

We agree that this is a drawback of our GTK dialog-based approach. Hence, we are planning an advanced mode of CoMA, where the user provides all settings already at the beginning of the run in the future. This might be done using a parameters file that can be handled over to the pipeline. This way, users do not have to type in their parameters continuously. Moreover, this advanced mode will facilitate the use of CoMA on HPC cluster systems, where jobs are typically scheduled.

2.3. I ran the pipeline strictly following the manual. It showed that I successfully ran the project. Although I saw several "Error" information in the terminal and the final log file, I thought they were just "warnings", such as "Error: (1431.1) FASTA-Reader: Warning: FASTA-Reader: ".

Thank you for pointing out this issue, this is interesting, as we have never encountered this warning message so far and neither of our several test candidates. Which CoMA version did you use (1.0 or 2.0)? Did you analyze the provided example dataset or your own data? After a check on StackOverflow, it seems as if there could be an issue with BLAST when using too long description lines (> 1000 characters). Maybe this was the case? Truncating the headers might solve the problem. We hope we were able to clarify this issue. We have also added this information in the CoMA manual; please see point 15 of chapter iv (Tutorial).

2.4. However, I tried some other choices and caused some errors such as Python package biom terminated by "UnboundLocalError: local variable data_start referenced before assignment". The CoMA pipeline didn't show me any useful information about whether it's a software problem or caused by a wrong parameter I chose. The funny thing is that the pipeline still showed "Process succeeded!" even though the error happened. In fact, I found the pipeline totally doesn't do exception handling. It's unacceptable.

In order to solve this issue, we would need to know at which step you received this error message (and ideally see the log file). Did you use the old version of CoMA? Most likely, the error message comes from a QIIME or Biom-Tools script that might be outdated. When using CoMA 2.0, none of our test candidates encountered such an error. Actually, CoMA does exception handling in various steps in order to prove user input parameters. However, errors originating from third party software (e.g. due to depreciated versions) cannot be caught in most cases. However, for such situations, we are providing a support e-mail service, where users can report their issues and we will fix each of them individually. There is also the plan to include a user forum to the CoMA web page.

3. I still think it's necessary to revise Figure 1 to show the main tools or parameters in each main step of the pipeline. In fact, I checked the source code of the pipeline in CoMA VM. Besides the python code for plotting and generating reports, there is not much code for the main steps of the pipeline. Therefore, it shouldn't be a problem to illustrate more detail of the main steps of the pipeline in one figure. And it's worth doing.

DONE. We have modified Figure 1 according to your suggestions and indicate now, where third party tools are used.

4. In fact, I agree with Reviewer #1's comment that the manuscript is too long and not focused. The authors should describe more details of the pipeline, such as the key parameters used in each step, instead of describing the real dataset.

DONE. Since two Reviewers criticized this point, we have decided to move the parts describing microbiological aspects of the soil samples to the supplementary material (S1 File). We are confident to better focus on the main purpose of this publication, which clearly is the presentation and evaluation of the CoMA pipeline. Thank you for this critical but important comment!

---

## [Decision Letter · Decision Letter 2]

4 Nov 2020

PONE-D-20-21940R2

CoMA – an intuitive and user-friendly pipeline for amplicon-sequencing data analysis

PLOS ONE

Dear Mr. Hupfauf,

Thank you for submitting your manuscript to PLOS ONE. After careful consideration, we feel that it has merit but does not fully meet PLOS ONE’s publication criteria as it currently stands. Therefore, we invite you to submit a revised version of the manuscript that addresses the points raised during the review process.

We look forward to receiving your revised manuscript.

Kind regards,

Zechen Chong

Academic Editor

PLOS ONE

Reviewers' comments:

Reviewer's Responses to Questions

**Comments to the Author**

1. If the authors have adequately addressed your comments raised in a previous round of review and you feel that this manuscript is now acceptable for publication, you may indicate that here to bypass the “Comments to the Author” section, enter your conflict of interest statement in the “Confidential to Editor” section, and submit your "Accept" recommendation.

Reviewer #2: (No Response)

2. Is the manuscript technically sound, and do the data support the conclusions?

Reviewer #2: Yes

3. Has the statistical analysis been performed appropriately and rigorously? 

Reviewer #2: Yes

4. Have the authors made all data underlying the findings in their manuscript fully available?

Reviewer #2: Yes

5. Is the manuscript presented in an intelligible fashion and written in standard English?

Reviewer #2: Yes

6. Review Comments to the Author

Reviewer #2: I am glad that the authors have made a lot changes to their software. I just tested the Singularity version of CoMA v2.0 in this review.

Major comment:

1. The authors responded some of my comments by asking me which version of COMA was used and telling me that I should use the latest v2.0 for testing. I used v1.0 because there was only v1.0 could be downloaded from their website before this round of review. I don't think I could test the current v2.0 before this round of revision. Now they have updated their website and added v2.0 download links.

2. The authors used old Singularity version (v2.x). They should clearly mention it in their manual because current Singularity 3.x is quite different from v2.x. The Singularity commands listed in their manual are unavailable in v3.x, such as "singularity image.create". In fact, I think it's better to use Singularity v3.x to build CoMA v2.0.

3. Singularity is good for HPC. As most of the HPCs are accessed by SSH, so the users need to set X11 forwarding to use the GUI from Singularity on remote HPC. Besides, add "-B ~/.Xauthority" parameter to command "singularity shell". It should be mentioned in their manual. Otherwise, the users might be not able to run the CoMA GUI correctly because it's not common to use GUI from Singularity. Besides, it might be more difficult to set this on Mac and Windows. The manual just say that it can be used on MacOS and Windows, but it didn't give any instructions.

4. There might be some problems in the step of renaming samples. For the tutorial, it only shows sample 2 and 3. Sample 1 was missing and it caused errors in downstream analysis. If I chose "Don't rename", then I can get results of downstream analysis.

Renaming of samples in progress ...

Please enter the new name for each sample:

2:

S2

3:

S3

Process succeeded! Your samples are renamed now!

5. For my previous comment point 2.4, I meant that CoMA cannot correctly check and handle potential error input of files or parameters. For example, if I input only one FASTQ file from the provided example files, and chose "paired end". It's definitely a naive mistake of user's input, but CoMA pipeline doesn't check this kind of mistakes and it continues. The main terminal was stuck at the "Trimming process proceeding ... " step and the size of detail log file increased with a lot of error message. It's just a very simple problem caused by parameter conflicts. I think CoMA should check the conflicts of parameters and give the users opportunity to re-input the parameters. In current CoMA v2.0, even I found that I just input a wrong parameter in previous step, I cannot change it and I have to stop and re-run the pipeline from the first step.

7. PLOS authors have the option to publish the peer review history of their article (what does this mean?). If published, this will include your full peer review and any attached files.

Reviewer #2: No

---

## [Author Response · Author response to Decision Letter 2]

12 Nov 2020

PONE-D-20-21940R2

Reviewer’s comments

We thank Reviewer #2 for his/her further suggestions. Below are the responses to the individual comments.

Reviewer #2:

I am glad that the authors have made a lot changes to their software. I just tested the Singularity version of CoMA v2.0 in this review.

Major comment:

1. The authors responded some of my comments by asking me which version of COMA was used and telling me that I should use the latest v2.0 for testing. I used v1.0 because there was only v1.0 could be downloaded from their website before this round of review. I don't think I could test the current v2.0 before this round of revision. Now they have updated their website and added v2.0 download links.

We would like to apologize for this confusion! Following the suggestions of another reviewer, we have provided CoMA 2.0 for download on our webpage several months ago. This download, however, required a user name and password, which we forgot to share with you. Sorry for that! We are glad that you could test the most current version of the tool now by using the Singularity image.

2. The authors used old Singularity version (v2.x). They should clearly mention it in their manual because current Singularity 3.x is quite different from v2.x. The Singularity commands listed in their manual are unavailable in v3.x, such as "singularity image.create". In fact, I think it's better to use Singularity v3.x to build CoMA v2.0.

DONE. We are aware of the big differences between Singularity v2.x and v3.x. Our intention was to guarantee most flexibility in usage. However, we agree with you that Singularity v3.x is most relevant nowadays and, hence, have decided to switch completely to this version. Since the creation of overlay images is no longer supported in the 3.x version and needs to be done manually by the users, we have decided to provide an already created overlay image file together with the Singularity image for download.

3. Singularity is good for HPC. As most of the HPCs are accessed by SSH, so the users need to set X11 forwarding to use the GUI from Singularity on remote HPC. Besides, add "-B ~/.Xauthority" parameter to command "singularity shell". It should be mentioned in their manual. Otherwise, the users might be not able to run the CoMA GUI correctly because it's not common to use GUI from Singularity. Besides, it might be more difficult to set this on Mac and Windows. The manual just say that it can be used on MacOS and Windows, but it didn't give any instructions.

DONE. Thank you for pointing out this important aspect! We have added a comment to the manual clarifying that X11 forwarding needs to be activated when using a SSH connection. This is indeed crucial since most HPC systems are accessed via SSH.

DONE. We believe that the additional option "-B ~/.Xauthority" is not necessary in most cases since Singularity binds the host system’s Home directory anyway when using the default settings of the tool. Nevertheless, we have added it now in order to guarantee functionality also in case of altered Singularity settings.

You are right; we are not explaining the usage of Singularity on Windows/MacOS in detail. We clearly encourage using VirtualBox for these operating systems, and Singularity (or the direct Linux installer) for Linux/HPC users. However, if anyone is still interested, we are providing links to the instruction page of Singularity (please see point ii.ii of the CoMA manual). The installation is explained here in detail and we do not believe that it is necessary to repeat it in the CoMA manual.

4. There might be some problems in the step of renaming samples. For the tutorial, it only shows sample 2 and 3. Sample 1 was missing and it caused errors in downstream analysis. If I chose "Don't rename", then I can get results of downstream analysis.

Renaming of samples in progress ...

Please enter the new name for each sample:

2:

S2

3:

S3

Process succeeded! Your samples are renamed now!

Thank you for reporting this. We double-checked the respective Python script and did not find any mistake that might cause such a behavior. In addition, neither of our test candidates reported a similar issue. We assume that your current OTU table file (“otu_table.txt”) might be corrupted for some reason. There is also the possibility that Sample 1 got lost in course of the subsampling process (a sample is discarded if the subsampling depth is higher than its read count). To clarify this, we would need to know how many samples are shown in the OTU-table before the renaming step and how many in the original OTU-table (which is the version directly created by LotuS).

5. For my previous comment point 2.4, I meant that CoMA cannot correctly check and handle potential error input of files or parameters. For example, if I input only one FASTQ file from the provided example files, and chose "paired end". It's definitely a naive mistake of user's input, but CoMA pipeline doesn't check this kind of mistakes and it continues. The main terminal was stuck at the "Trimming process proceeding ... " step and the size of detail log file increased with a lot of error message. It's just a very simple problem caused by parameter conflicts. I think CoMA should check the conflicts of parameters and give the users opportunity to re-input the parameters. In current CoMA v2.0, even I found that I just input a wrong parameter in previous step, I cannot change it and I have to stop and re-run the pipeline from the first step.

DONE. We agree that not checking for correctly assigned input files was problematic. Hence, we have decided to implement a step that is checking the user input during sample registration as well as the provided input files. CoMA now warns the user and terminates the run, if the common part of the sequences was not entered correctly as well as if paired-end samples are missing. Moreover, CoMA catches now errors evoked by missing files in multiple situations and prints it in the terminal window. The error messages are short and clear in order to help the user to solve the issue immediately. Thank you for these important inputs; they truly improved the functionality of the pipeline!

---

## [Editor Report · Decision Letter 3]

18 Nov 2020

CoMA – an intuitive and user-friendly pipeline for amplicon-sequencing data analysis

PONE-D-20-21940R3

Dear Sebastian Hupfauf, 

We’re pleased to inform you that your manuscript has been judged scientifically suitable for publication and will be formally accepted for publication once it meets all outstanding technical requirements.

Kind regards,

Zechen Chong

Academic Editor

PLOS ONE
---

## [Editor Report · Acceptance letter]

20 Nov 2020

PONE-D-20-21940R3 

CoMA – an intuitive and user-friendly pipeline for amplicon-sequencing data analysis 

Dear Dr. Hupfauf:

I'm pleased to inform you that your manuscript has been deemed suitable for publication in PLOS ONE. Congratulations! Your manuscript is now with our production department. 

Kind regards, 

on behalf of

Dr. Zechen Chong 

Academic Editor

PLOS ONE